# A stochastic framework of neurogenesis underlies the assembly of neocortical cytoarchitecture

Alfredo Llorca[1,2], Gabriele Ciceri[1,2†], Robert Beattie[3], Fong Kuan Wong[1,2], Giovanni Diana[1,2], Eleni Serafeimidou-Pouliou[1,2], Marian Fernández-Otero[1,2], Carmen Streicher[3], Sebastian J Arnold[4], Martin Meyer[1,2], Simon Hippenmeyer[3], Miguel Maravall[5], Oscar Marin[1,2]*

[1]Centre for Developmental Neurobiology, Institute of Psychiatry, Psychology and Neuroscience, King's College London, London, United Kingdom; [2]MRC Centre for Neurodevelopmental Disorders, King's College London, London, United Kingdom; [3]Institute of Science and Technology Austria, Klosterneuburg, Austria; [4]Institute of Experimental and Clinical Pharmacology and Toxicology, Faculty of Medicine, University of Freiburg, Freiburg, Germany; [5]Sussex Neuroscience, School of Life Sciences, University of Sussex, Brighton, United Kingdom

**Abstract** The cerebral cortex contains multiple areas with distinctive cytoarchitectonic patterns, but the cellular mechanisms underlying the emergence of this diversity remain unclear. Here, we have investigated the neuronal output of individual progenitor cells in the developing mouse neocortex using a combination of methods that together circumvent the biases and limitations of individual approaches. Our experimental results indicate that progenitor cells generate pyramidal cell lineages with a wide range of sizes and laminar configurations. Mathematical modeling indicates that these outcomes are compatible with a stochastic model of cortical neurogenesis in which progenitor cells undergo a series of probabilistic decisions that lead to the specification of very heterogeneous progenies. Our findings support a mechanism for cortical neurogenesis whose flexibility would make it capable to generate the diverse cytoarchitectures that characterize distinct neocortical areas.

*For correspondence:
oscar.marin@kcl.ac.uk

Present address: †Center for Stem Cell Biology, Memorial Sloan-Kettering Cancer Center, New York, United States

Competing interests: The authors declare that no competing interests exist.

## Introduction

The mammalian cerebral cortex contains a wide diversity of neuronal types heterogeneously distributed across layers and regions. The most abundant class of neurons in the cerebral cortex are excitatory projection neurons, also known as pyramidal cells (PCs). In the neocortex, PCs can be further classified into several subclasses with unique laminar distributions, projection patterns and electrophysiological properties (*Greig et al., 2013*; *Jabaudon, 2017*; *Lodato and Arlotta, 2015*), and currently available data suggest that several dozen distinct transcriptional signatures can be distinguished among them (*Tasic et al., 2018*). The relative abundance of the different types of PCs largely determines the distinct cytoarchitectonical patterns observed across different regions of the mammalian neocortex (*Brodmann and Gary, 2006*).

The diversity of excitatory neurons emerges from progenitor cells in the ventricular zone (VZ) of the developing neocortex known as radial glial cells (RGCs) (*Malatesta et al., 2000*; *Miyata et al., 2001*; *Noctor et al., 2001*). RGCs divide symmetrically to expand the progenitor pool during early stages of corticogenesis. Subsequently, they undergo asymmetric cell divisions to generate clones of PCs directly or indirectly via intermediate progenitor cells (IPCs) (*Lui et al., 2011*; *Taverna et al., 2014*). The characteristic vertical organization of migrating PCs in the developing neocortex led to

**eLife digest** Recognizable by its deep outer folds in humans, the cerebral cortex is a region of the mammalian brain which handles complex processes such as conscious perception or decision-making. It is organized in several layers that contain different types of 'excitatory' neurons which can activate other cells. The various areas of the cortex have different characteristics as they contain various proportions of each kind of neurons.

Stem cells are cells capable to divide and create various types of specialized cells. The excitatory neurons in the cortex are created during development by stem cells known as radial glial cells. These cells divide several times, giving rise to different types of neurons in sucessive divisions, presumably thanks to internal molecular clocks. In the cortex, it is generally assumed that an individual radial glial cell produces all the different types of excitatory neurons. However, studies have suggested that certain cells could be specialized in creating specific types of neurons.

To explore this question, Llorca et al. used three complementary approaches to follow individual radial glial cells and track the neurons they created in mouse embryos. This helped to understand how groups of stem cells work together to build the cortex. The experiments revealed that radial glial cells differ more than anticipated in the number and the types of neurons they generate, and rarely produce all types of excitatory neurons. In other words, the output of individual radial glial cells is not always the same. The results by Llorca et al. suggest that as radial glial cells divide, they undergo a series of probabilistic decisions – that is, in each division the cells have a certain probability to generate a specific type of neuron. Consequently, the resulting lineages are rarely identical or contain all types of excitatory neurons, but collectively they generate the full diversity of excitatory neurons in the cortex. Ultimately, new insights into how excitatory neurons form and connect in the brain may be used to help understand psychiatric conditions where circuits in the cortex might be impaired, such as in autism spectrum disorders.

the 'radial unit hypothesis', which postulates that PCs in a given radial column are clonally related (*Rakic, 1988*). However, the precise mechanisms through which RGCs generate diverse cytoarchitectonic patterns throughout the neocortex remain to be elucidated.

The most commonly accepted view of cortical neurogenesis is based on the notion that RGCs are multipotent and generate all types of excitatory neurons following an exquisite inside-out temporal sequence (*Leone et al., 2008*; *Molyneaux et al., 2007*; *Rakic et al., 1994*). Consistently, progenitor cells cultured in vitro reproduce the temporal sequence of cortical neurogenesis (*Gaspard et al., 2008*; *Shen et al., 2006*), and genetic fate mapping experiments have shown that cortical progenitors identified by the expression of the transcription factors Fezf2 and Sox9 are multipotent in vivo (*Guo et al., 2013*; *Kaplan et al., 2017*). In contrast to this view, other studies have suggested the existence of fate-restricted cortical progenitors, which would only generate PCs for certain layers of the neocortex (*Franco et al., 2012*; *García-Moreno and Molnár, 2015*). However, the interpretation of these results remains a matter of controversy (*Eckler et al., 2015*; *Gil-Sanz et al., 2015*).

Our current framework for understanding cortical neurogenesis largely relies on studies that consider RGCs as a homogeneous population. Consistent with this view, recent clonal analyses of the developing neocortex led to the conclusion that progenitor cell behavior conforms to a deterministic program through which individual RGCs consistently generate the same neuronal output (*Gao et al., 2014*). This would suggest that variations in the organization of cortical areas would exclusively rely on mechanisms of lineage refinement at postmitotic stages, such as programmed cell death. Alternatively, the absence of detailed quantitated data of individual PC lineages or methodological caveats may have prevented the identification of a certain degree of heterogeneity in the neuronal output of individual RGCs.

In this study, we have used three complementary approaches to circumvent some of the intrinsic technical biases associated with each of the previously used methods to systematically investigate the clonal organization of PC lineages in the cerebral cortex. Our results provide a detailed quantitative assessment of the neurogenic fate of individual VZ progenitor cells that reveal a large diversity of PC lineage configurations. These findings support a stochastic model of cortical neurogenesis

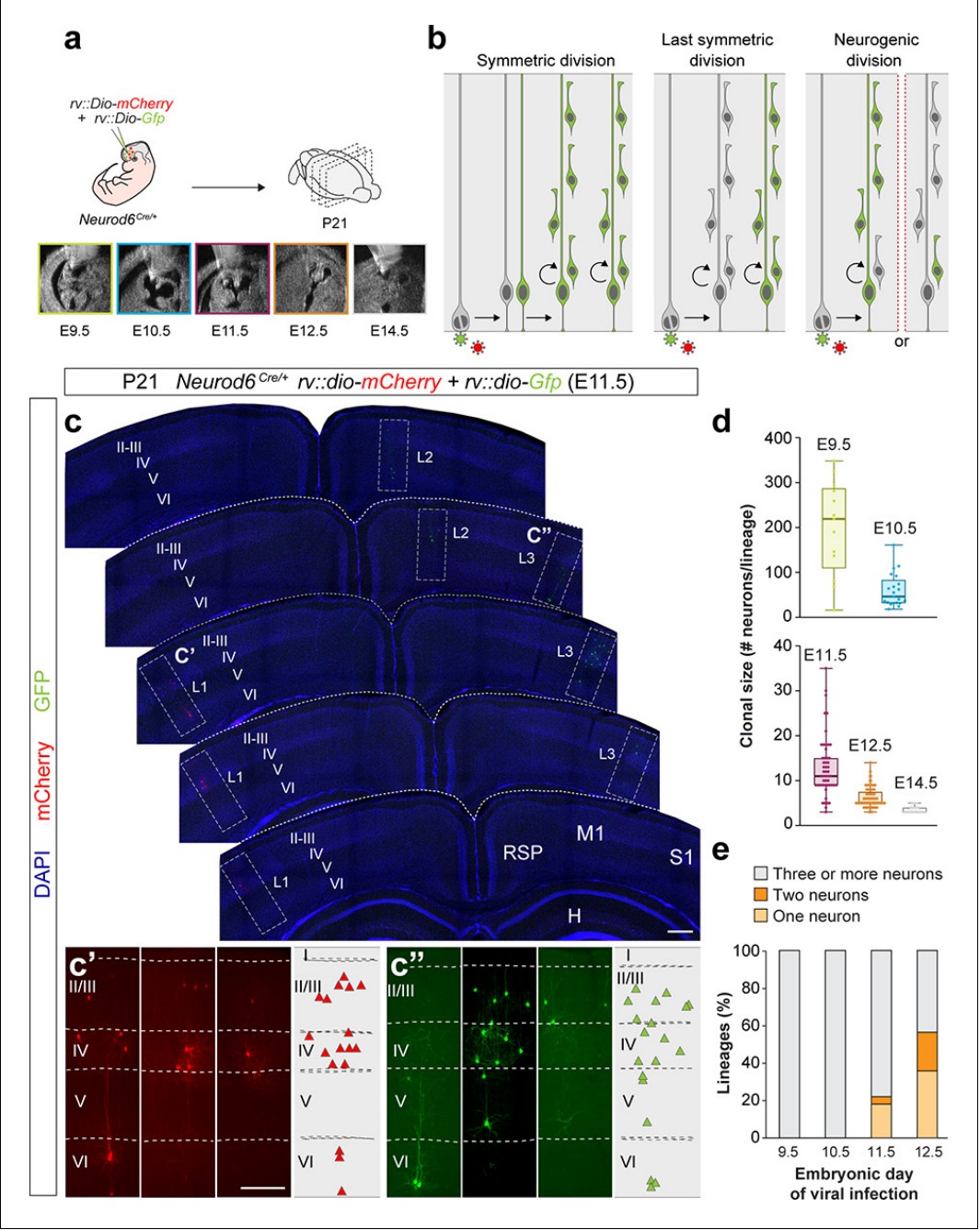

**Figure 1.** Identification of pyramidal cell lineages with low-titer conditional reporter retroviruses. (a) Experimental paradigm. (b) Schematic representation of the expected labeling outcomes in retroviral lineage tracing experiments. (c–c') Serial 100 μm coronal sections through the telencephalon of a P21 *Neurod6^Cre/+* mouse infected with low-titer conditional reporter retroviruses at E11.5. Lineages (L) 1 and 3 are shown at high magnification in c' and c'', respectively. Dashed lines define external brain boundaries and cortical layers. The schemas collapse lineages spanning across several sections into a single diagram. (d) Quantification of the number of PCs per lineage in P21 *Neurod6^Cre/+* mice infected with conditional reporter retroviruses at different embryonic stages. Lineages smaller than three cells were excluded. Boxes show median and inter-quartile distance, whiskers correspond to minimum and maximum values. Colored dots show individual clonal size values. (e) Quantification of the fraction of cortical lineages containing one, two or three or more neurons in P21 *Neurod6^Cre/+* mice infected with conditional reporter retroviruses at different embryonic stages. n = 13 lineages in three animals at E9.5; 21 lineages in three animals at E10.5; 64 lineages in five animals at E11.5; 166 lineages in seven animals at E12.5; 32 lineages in four animals at E14.5. I–VI, cortical layers I to VI; H, hippocampus area; M1, primary motor cortex; RSD,

*Figure 1 continued*

retrosplenial cortex; S1, primary somatosensory cortex. Scale bars equal 100 μm (c) and 300 μm (c' and c"). Data used for quantitative analyses as well as the numerical data that are represented in graphs are available in *Figure 1—source data 1* and *Figure 1—source data 2*. See also *Figure 1—figure supplement 1*.

The online version of this article includes the following source data and figure supplement(s) for figure 1:

**Source data 1.** E9.5-E14.5 *Neurod6$^{Cre/+}$* retroviral lineages analyzed at P21.

**Source data 2.** Summary of numerical data that are represented in graphs.

**Figure supplement 1.** Sparse labeling of neuronal clones with low-titer retroviral infection.

through which a limited number of progenitor cell identities could generate the diverse of cytoarchitectonical patterns observed in the neocortex.

## Results

### Retroviral tracing of pyramidal cell lineages

To study the cellular mechanisms underlying the generation of PCs in the neocortex, we analyzed the output and organization of neuronal lineages generated by individual progenitor cells. To this end, we first used replication-deficient retroviral vectors that integrate indiscriminately in mitotic cells but only identify cell lineages with fluorescent proteins following Cre-dependent recombination (*Ciceri et al., 2013*). To specifically label PC lineages, we injected a very low titer cocktail of conditional reporter retroviruses (*rv::dio-Gfp* and *rv::dio-mCherry*) into the lateral ventricle of *Neurod6-$^{Cre/+}$* mouse embryos (also known as *Nex-Cre*), in which Cre expression is confined to postmitotic PCs (*Goebbels et al., 2006*) (*Figure 1a*). Using this approach, we achieved sparse labeling and avoided biasing the tagging of progenitor cells by the expression of specific genetic markers (*Cepko et al., 2000*).

To identify the developmental stage at which progenitor cells become neurogenic in the cortex, we injected retroviruses at different embryonic days (E9.5 to E14.5) and analyzed the organization of individual PC clusters at postnatal day (P) 21 (*Figure 1a*). Since a single copy of the viral vector is stably integrated into the host genome, retroviral infection leads to the labeling of only one of the two daughter cells resulting from the division of the infected progenitor cell. Consequently, infection of progenitor cells in the ventricular zone (VZ) of the pallium labels PC lineages in three main configurations depending of the mode of division of the infected progenitor (*Figure 1b*): (1) a large cluster containing more than one lineage, which results from the infection of a self-renewing progenitor cell dividing symmetrically; (2) a single lineage, which results from the infection of a progenitor cell undergoing its last symmetric division; and (3) a partial lineage, which results from a neurogenic division of a progenitor cell. In this later case, partial lineages may contain the majority of neurons in the clone, if integration occurs in the progenitor cell, or one or two neurons, if the integration occurs in a neuron or an IPC, respectively.

We observed clusters of neurons with the characteristic morphology of PCs at all stages examined. Systematic mapping at P21 revealed very sparse labeling and widespread distribution of clones throughout the entire neocortex (*Figure 1c–c"* and *Figure 1—figure supplement 1*). The spatial segregation of the lineages was confirmed by the virtual absence of green and red clones within 500 μm of each other in all experiments analyzed (*Figure 1—figure supplement 1*). We quantified the number of PCs per clone at P21 following viral infection at different embryonic stages and observed that lineages contain progressively smaller progenies (*Figure 1d*). This is consistent with the notion that VZ progenitors undergo proliferative symmetric cell divisions early during corticogenesis before they become neurogenic and begin self-renewing via asymmetric divisions (*Götz and Huttner, 2005*; *Kriegstein and Götz, 2003*). Since neurogenic divisions label one or two neurons in 50% of the cases (*Figure 1b*), the fraction of one- and two-cell clones found after retroviral infection is indicative of the proportion of neurogenic VZ progenitor cells at each embryonic stage. We observed that these clones represent ~50% of the lineages at E12.5 (*Figure 1e*). Consistent with previous reports using other methods (*Gao et al., 2014*), these results indicated that the onset of cortical neurogenesis begins immediately before E12.5, and that at this stage most VZ progenitor cells are already neurogenic. Thus, we focused subsequent analyses on this stage.

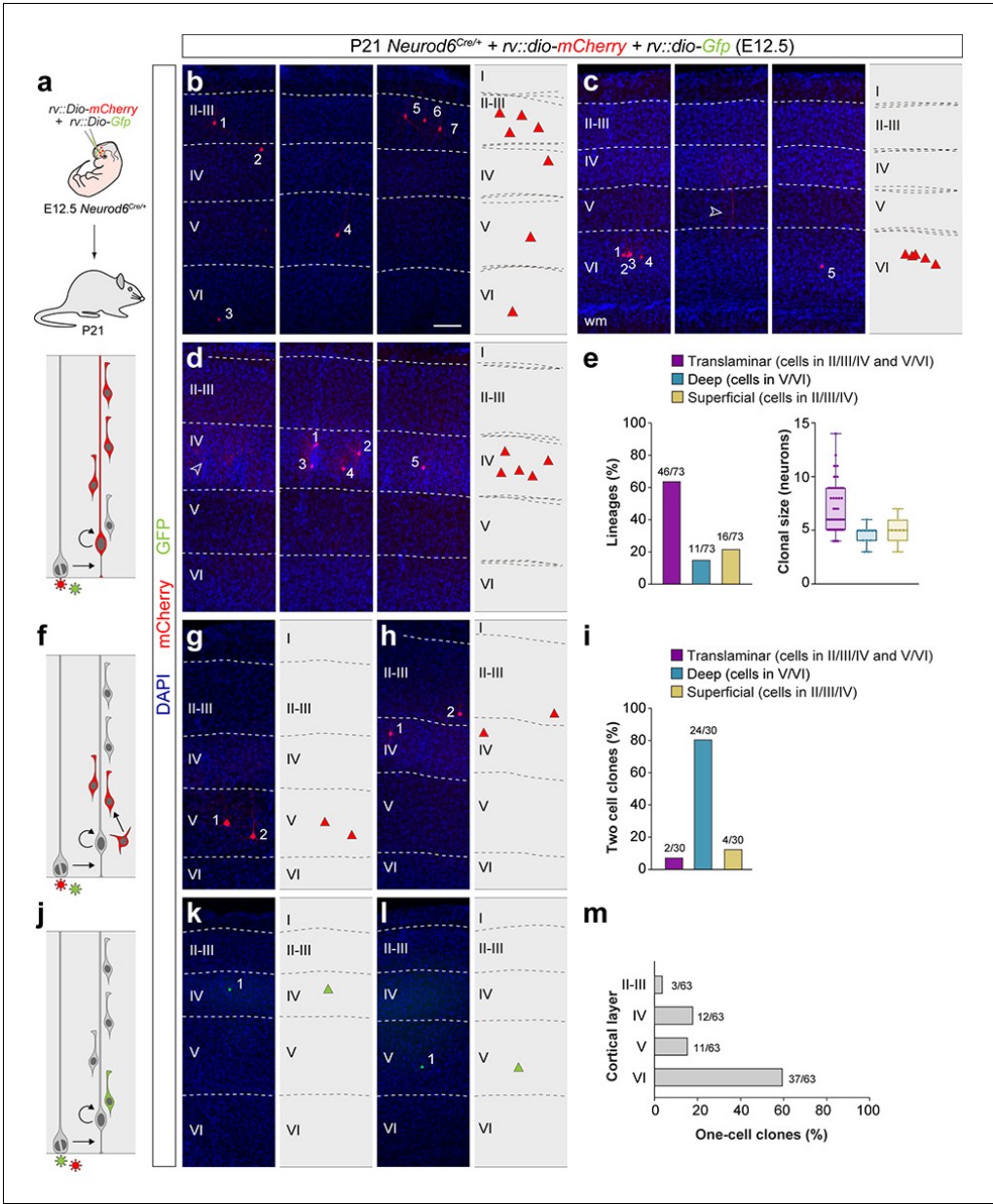

**Figure 2.** Retroviral-based lineage tracing reveals diverse lineage outcomes. (**a**) Experimental paradigm. The bottom panel illustrates the expected labeling outcome following retroviral infection of an RGC undergoing a neurogenic cell division in which the viral integration occurs in the self-renewing RGC. (**b–d**) Serial 100 µm coronal sections through the cortex of P21 *Neurod6*$^{Cre/+}$ mice infected with low-titer conditional reporter retroviruses at E12.5. The images show examples of translaminar (**b**), deep-layer restricted (**c**) and superficial-layer restricted (**d**) lineages containing three or more cells. Dashed lines define external brain boundaries and cortical layers. The schemas collapse lineages spanning across several sections into a single diagram. Example images illustrating each lineage correspond to sequential sections of the same brain. (**e**) Quantification of the fraction of translaminar, deep- and superficial-layer restricted lineages containing three or more cells, and clonal size. Boxes show median and inter-quartile distance, whiskers correspond to minimum and maximum values. Colored dots show individual clonal size values. (**f**) Expected labeling outcome following retroviral infection of an RGC undergoing a neurogenic cell division in which the viral integration occurs in an IPC (indirect neurogenesis). (**g,h**) Coronal sections through the cortex of P21 *Neurod6*$^{Cre/+}$ mice infected with low-titer conditional reporter retroviruses at E12.5. The images show examples of superficial and deep layer-restricted two-cell clones. (**i**) Quantification of the fraction of translaminar, deep and superficial layer-restricted two-cell lineages. (**j**) Expected labeling outcome following retroviral infection of an RGC undergoing a neurogenic cell division in which the viral integration occurs in a postmitotic neuron (direct neurogenesis). (**k,l**) Coronal sections through the cortex of P21 *Neurod6*$^{Cre/+}$ mice

*Figure 2 continued on next page*

*Figure 2 continued*

infected with low-titer conditional reporter retroviruses at E12.5. The images show examples of superficial and deep layer-restricted single-cell clones. (m) Laminar distribution of single-cell clones. n = 166 lineages in seven animals. Scale bar equals 100 μm. Data used for quantitative analyses as well as the numerical data that are represented in graphs are available in *Figure 2—source data 1* and *Figure 1—source data 2*. See also *Figure 2—figure supplement 1*.

The online version of this article includes the following source data and figure supplement(s) for figure 2:

**Source data 1.** E12.5 *Neurod6*$^{Cre/+}$ retroviral lineages analyzed at P21.
**Figure supplement 1.** Lineage tracing of Tbr2+ intermediate progenitor cells.
**Figure supplement 1—source data 1.** E12.5 *Tbr2*$^{CreERT2/+}$;*RCL-Gfp* lineages analyzed at P21.

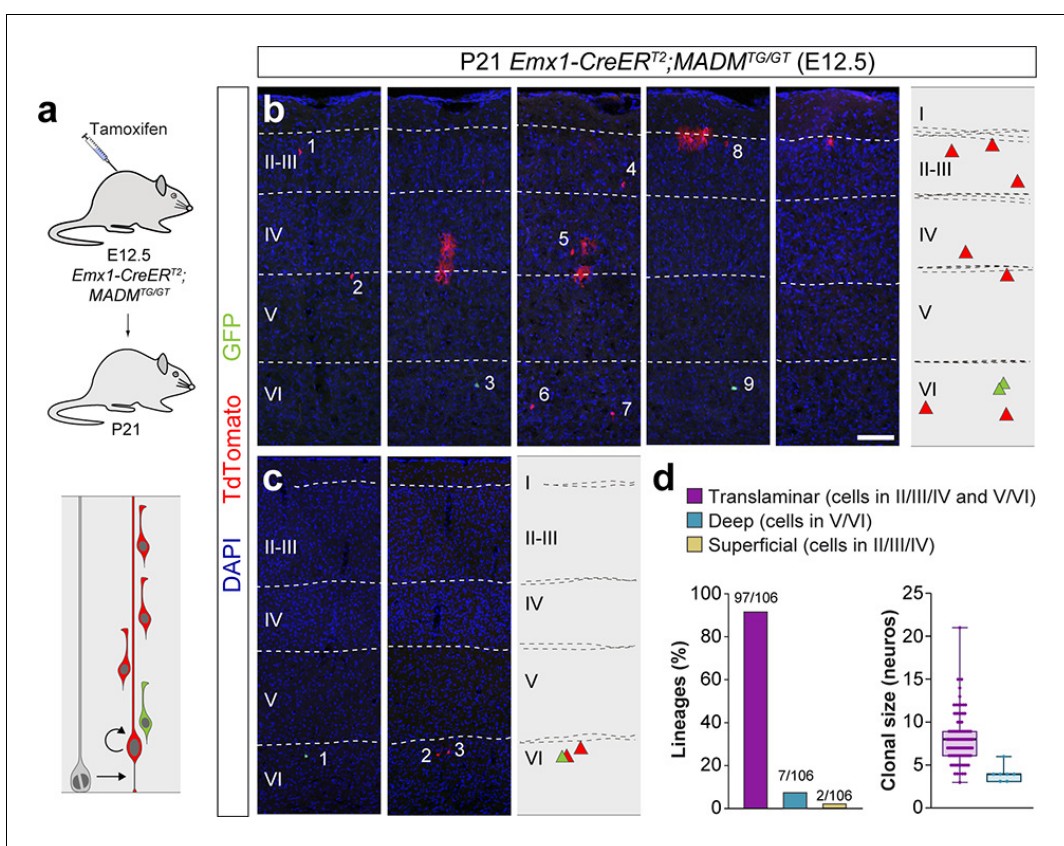

**Figure 3.** Lineage tracing using MADM identifies a small fraction of deep layer-restricted cortical lineages. (a) Experimental paradigm. The bottom panel illustrates the expected labeling outcome of a neurogenic RGC division following inducible MADM-based lineage tracing in which two subclones are labeled with different reporters. (b,c) Serial 100 μm coronal sections through the cortex of P21 *Emx1-CreER*$^{T2}$;MADM$^{TG/GT}$ mice treated with tamoxifen at E12.5. The images show examples of translaminar (b) and deep layer-restricted (c) lineages. The schemas collapse lineages spanning across several sections into a single diagram. Example images illustrating each lineage correspond to sequential sections of the same brain. (d) Quantification of the fraction of translaminar, deep and superficial layer-restricted lineages, and clonal size in MADM lineages derived from a neurogenic (asymmetric) RGC division. Boxes show median and inter-quartile distance, whiskers correspond to minimum and maximum values. Colored dots show individual clonal size values. n = 106 neurogenic lineages in 28 animals. Scale bar equals 100 μm. Data used for quantitative analyses as well as the numerical data that are represented in graphs are available in *Figure 3—source data 1* and *Figure 1—source data 2*. See also *Figure 3—figure supplement 1*.

The online version of this article includes the following source data and figure supplement(s) for figure 3:

**Source data 1.** E12.5 *Emx1-CreER*$^{T2}$;MADM$^{TG/GT}$ lineages analyzed at P21.
**Figure supplement 1.** Large MADM subclones reveal a small fraction of artifactual superficial layer-restricted lineages in the retroviral experiments.

## Cortical progenitors exhibit heterogeneous neuronal output

We first examined lineages labeled at E12.5 that contained more than two cells, which correspond to the progeny of a VZ progenitor cell (*Figure 2a*). Consistent with classical models of cortical neurogenesis, we found that most VZ progenitor cells (63%) infected with retroviruses at E12.5 produce translaminar lineages containing neurons in both deep (V and VI) and superficial (II-III and IV) layers of the neocortex (*Figure 2b,e*). However, we also observed a substantial fraction of lineages in which PCs were confined to either deep (*Figure 2c,e*) or superficial (*Figure 2d,e*) layers (15% and 22%, respectively).

The distribution of single-cell and two-cell clones following infection of VZ progenitor cells at E12.5 further support the existence of cortical lineages restricted to superficial layers of the neocortex. As expected from the normal progression of neurogenesis in translaminar lineages, most single-

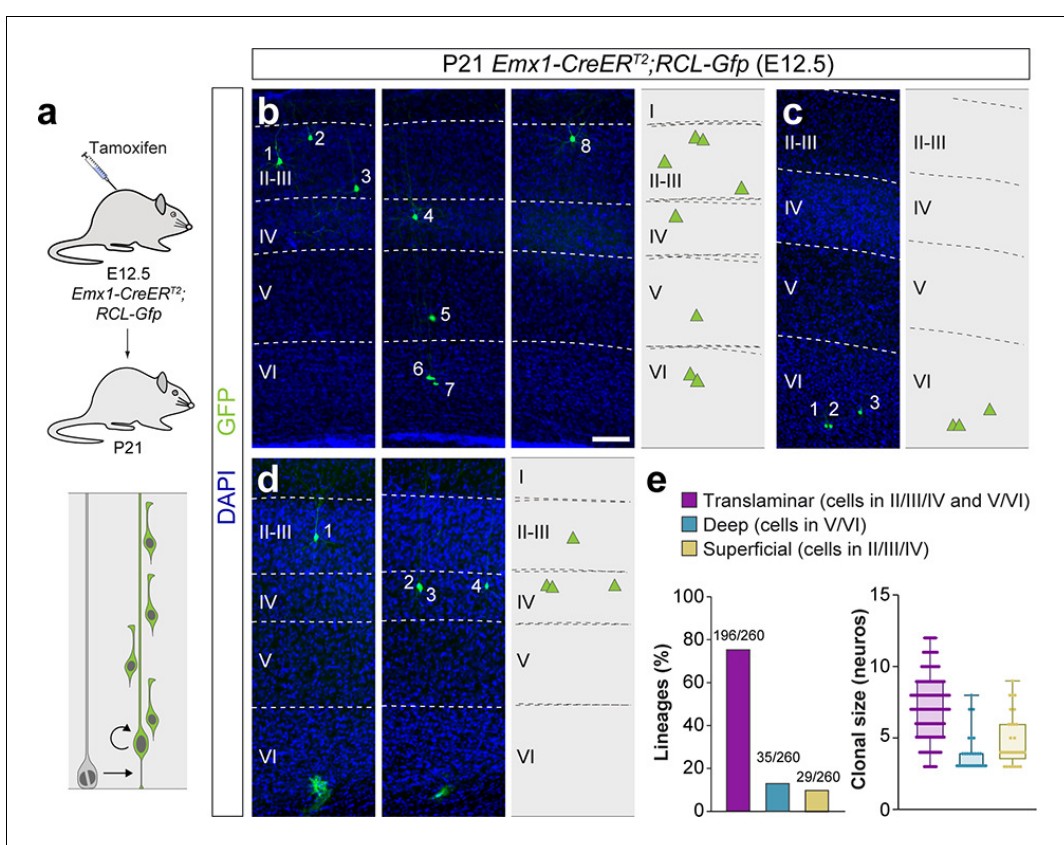

**Figure 4.** A fraction of early-quiescent cortical progenitors generates superficial layer-restricted lineages. (a) Experimental paradigm. The bottom panel illustrates the expected labeling outcome of a neurogenic RGC division following inducible conditional reporter lineage tracing in *Emx1-CreER^T2^;RCL-Gfp* mice. (b–d) Serial 100 μm coronal sections through the cortex of P21 *Emx1-CreER^T2^;RCL-Gfp* mice treated with low-dose tamoxifen at E12.5. The images show examples of translaminar (b), deep layer-restricted (c) and superficial layer-restricted (d) lineages. The schemas collapse lineages spanning across several sections in a single diagram. Example images illustrating each lineage correspond to sequential sections of the same brain. (e) Quantification of the fraction of translaminar, deep and superficial layer-restricted lineages, and clonal size in inducible conditional reporter lineage-tracing experiments. Boxes show median and inter-quartile distance, whiskers correspond to minimum and maximum values. Colored dots show individual clonal size values. n = 260 neurogenic lineages in 25 animals. Scale bar equals 100 μm. Data used for quantitative analyses as well as the numerical data that are represented in graphs are available in *Figure 4—source data 1* and *Figure 1—source data 2*. See also *Figure 4—figure supplement 1*.

The online version of this article includes the following source data and figure supplement(s) for figure 4:

**Source data 1.** E12.5 *Emx1-CreER^T2^;RCL-Gfp* lineages analyzed at P21.

**Figure supplement 1.** Superficial layer-restricted lineages in the murine cerebral cortex.

cell and two-cell clones (which result from the labeling of a neuron or an IPC, respectively) were located in deep layers of the cortex (*Figure 2f–m*). However, in these experiments, we also identified a small fraction of single-cell and two-cell clones in superficial layers of the neocortex (*Figure 2f–m*). This suggested that some VZ progenitor cells generate PCs for superficial layers of the neocortex in their earliest neurogenic divisions.

We noticed that the clonal size of laminar-restricted lineages is typically smaller than that of translaminar clones (*Figure 2e*). One explanation for this difference could be that laminar-restricted lineages represent sub-clones resulting from the labeling of IPCs that undergo more than one round of cell division, generating four to five neurons with a laminar-restricted distribution. To test this hypothesis, we carried out lineage-tracing experiments at single-cell resolution using low-dose tamoxifen administration in *Tbr2$^{CreERT2/+}$;RCL-Gfp* pregnant mice at E12.5 (*Figure 2—figure supplement 1a*), which led to the sparse labeling of IPCs and their progenies (*Pimeisl et al., 2013*). We analyzed 73 IPC-derived lineages at P21 and exclusively found one-cell and two-cell clones, with no evidence for larger clones within our sample (*Figure 2—figure supplement 1b–d*). Although the existence of IPCs that undergo more than one cell division cannot be completely excluded, these results indicated that this is not common at this developmental stage. Consequently, IPCs are unlikely to be the origin of laminar-restricted lineages.

## A small fraction of progenitors generates laminar-restricted lineages

Our retroviral tracing experiments suggested that the neuronal output of neocortical VZ progenitor cells is significantly more heterogeneous than previously described, including translaminar, deep- and superficial-layer restricted lineages. However, several technical limitations may contribute to the observation of laminar-restricted lineages, as retroviral tracing may lead to the incomplete labeling of neuronal lineages. For example, the existence of deep layer-restricted lineages might be due to the silencing of the viral cassette after a few rounds of cell division (*Cepko et al., 2000*), which would prevent the expression of GFP or mCherry in superficial layer PCs. There are also alternative explanations for the observation of superficial layer-restricted lineages in the retroviral-tracing experiments. First, infected progenitors might have become neurogenic at slightly earlier stages and have already produced a wave of deep layer PCs before infection, which would therefore not be labeled by the retrovirus. Second, the entire set of deep layer neurons might have been generated during the first neurogenic division of a VZ progenitor cell, which would not be labeled in some cases due to the retroviral integration mechanism.

To overcome these technical limitations, we took advantage of the Mosaic Analysis with Double Markers (MADM) technique, a genetic method widely used to fate-map cellular lineages at high resolution (*Hippenmeyer et al., 2010*; *Zong et al., 2005*). We used the *Emx1-CreER$^{T2}$* mice (*Kessaris et al., 2006*) to induce MADM sparse labeling of VZ progenitor cells following tamoxifen administration at E12.5 (*Figure 3a*). We specifically focused our analysis on G2-X MADM segregation events that result in the labeling of an unbalanced number of daughter cells with either green or red fluorescent proteins and report the outcome of asymmetric divisions in VZ progenitor cells (*Zong et al., 2005*). Consistent with the retroviral lineage tracing experiments, we found that the vast majority of MADM lineages adopt a translaminar configuration (*Figure 3b,d*). In addition, we also identified some lineages in which PCs were confined to layers V and VI, thereby confirming the existence of cortical lineages restricted to deep layers of the neocortex (*Figure 3c,d*). We observed that the fraction of deep layer-restricted lineages labeled with MADM (~7%) is smaller than that obtained with retroviral tracing (15%), which suggested that reporter silencing might exist in some clones in the retroviral lineage tracing experiments. In contrast, we did not recover a significant number of superficial layer-restricted lineages in MADM experiments (*Figure 3d*).

The MADM experiments suggested that the observation of superficial layer-restricted lineages in retroviral experiments might be artifactual, a result of the incomplete retroviral labeling of neuronal lineages. We reasoned that if this were the case, the analysis of the MADM sub-clones (i.e. only one of the two colors in the lineage) containing more than two cells should lead to a similar fraction of 'artifactual' lineages, since these would essentially correspond to those labeled by retroviral infection missing the first division of VZ progenitor cells (*Figure 3—figure supplement 1*). This analysis indeed identified a small fraction of MADM sub-clones as 'apparent' superficial layer-restricted lineages (~12%), which was nevertheless significantly smaller than those identified in the retroviral

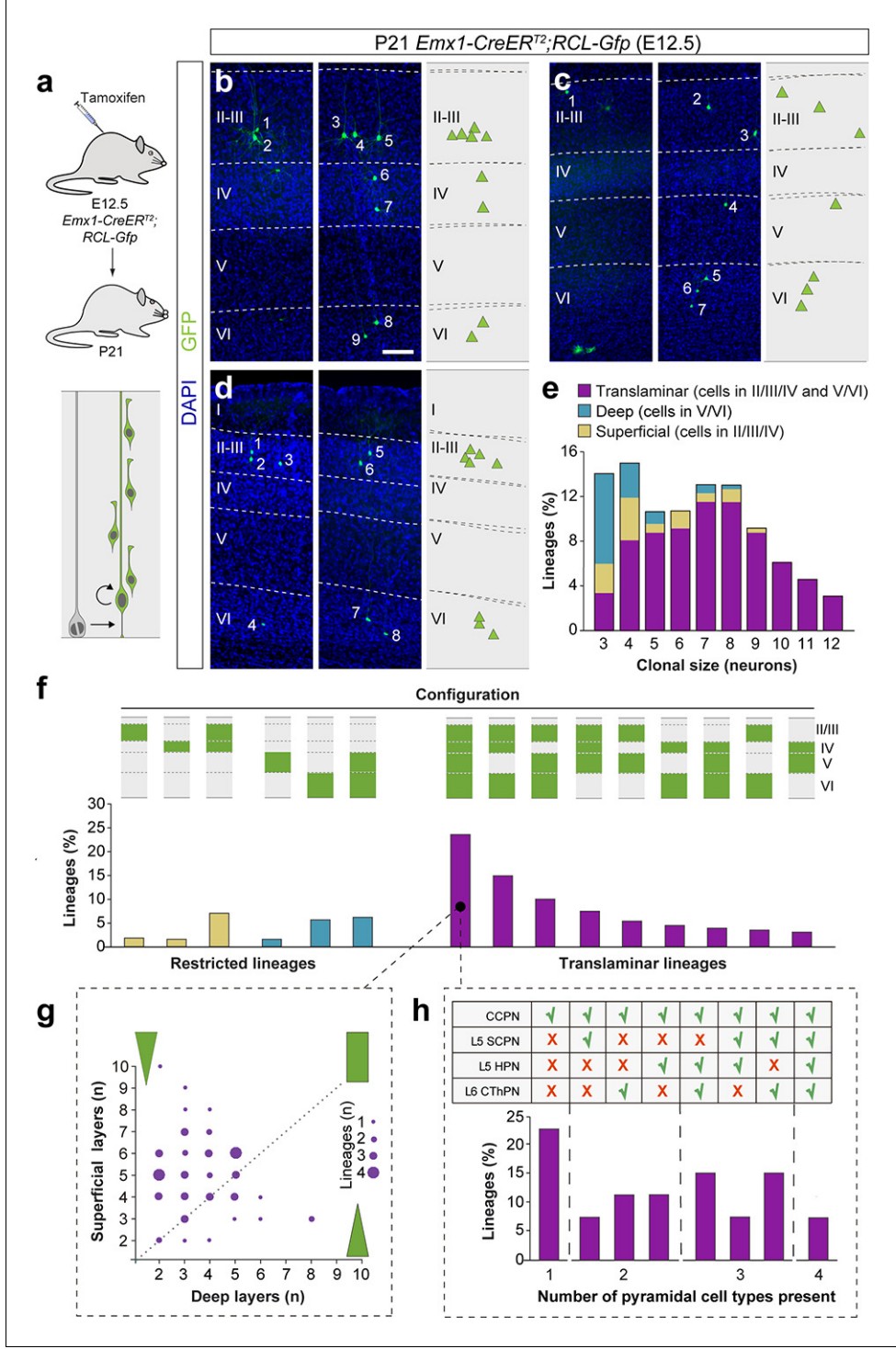

**Figure 5.** Translaminar lineages adopt very heterogeneous configurations. (**a**) Experimental paradigm. The bottom panel illustrates the expected labeling outcome of a neurogenic RGC division following inducible conditional reporter lineage tracing in *Emx1-CreER^T2;RCL-Gfp* mice. (**b–d**) Serial 100 μm coronal sections through the cortex of P21 *Emx1-CreER^T2;RCL-Gfp* mice treated with low-dose tamoxifen at E12.5. The images show examples of translaminar lineages with various laminar configurations observed in the somatosensory (**b and c**) and visual cortex (**d**).The schemas collapse lineages spanning across several sections into a single diagram. Example images illustrating each lineage correspond to sequential sections of the same brain. (**e**) Clonal size distribution of translaminar, deep and superficial layer-restricted lineages. (**f**) Relative frequency (expressed as percentage over the total number of lineages) of the different laminar configurations (green and gray schemas) in inducible

*Figure 5 continued on next page*

*Figure 5 continued*

conditional reporter lineage tracing experiments. (**g**) Relative abundance of PCs in superficial and deep layers from translaminar lineages containing cells in every layer. Lineages are represented as circles in a bi-dimensional space, indicating the number of cells in superficial versus deep layers. The size of the circle indicates the number of lineages that shown a particular configuration. Green shapes schematically represent lineage configurations. A rectangular shape illustrates lineages with a balanced number of superficial and deep PCs; triangular shapes represent configurations of lineages biased towards superficial or deep layer neurons. (**h**) Fraction of translaminar lineages with neurons in every layer containing one, two, three or four subclasses of PCs. n = 260 neurogenic lineages in 25 animals. CCPN, cortico-cortical projection neuron; SCPN, subcortical projection neuron; HPN, heterogeneous projection neuron; CThPN, cortico-thalamic projection neuron. Scale bar equals 100 μm. Data used for quantitative analyses as well as the numerical data that are represented in graphs are available in *Figure 4—source data 1*, *Figure 5—source data 1* and *Figure 1—source data 2*. See also *Figure 5—figure supplements 1–3*.

The online version of this article includes the following source data and figure supplement(s) for figure 5:

**Source data 1.** Projection neuron markers in E12.5 *Emx1-CreER^{T2};RCL-Gfp* all-layer lineages analyzed at P21.
**Figure supplement 1.** Retrovirus and MADM labeled lineages exhibit a diversity of laminar configurations.
**Figure supplement 2.** Identification of pyramidal cell subclasses.
**Figure supplement 2—source data 1.** Projection neuron marker colocalization.
**Figure supplement 3.** Subtle impact of pyramidal neuron cell death in final configurations of cortical neuron lineages.
**Figure supplement 3—source data 1.** E12.5 *Emx1-CreER^{T2};RCL-Gfp* lineages analyzed at P2.

experiments (22%). This indicated that although some of the superficial layer-restricted lineages observed in retroviral experiments were artifactual, others might not be.

One important difference between both approaches is that MADM G2-X recombination events occur exclusively in mitotic cells (*Zong et al., 2005*), while retroviral labeling does not strictly depend on cell division. Retroviruses require cell division for their integration into the genome, but the infection is independent of cell cycle stage (*Cepko et al., 2000*). Thus, we hypothesized that MADM may not consistently label a fraction of quiescent or slowly dividing progenitors, which could otherwise be targeted by retroviral infection. To test this idea, we carried out a new set of lineage-tracing experiments using a third, complementary method. In brief, we traced cortical lineages at single cell resolution using low-dose tamoxifen administration in *Emx1-CreER^{T2};RCL-Gfp* (*RCL-Gfp* also known as RCE) pregnant mice at E12.5 (*Figure 4a*), in which labeling of VZ progenitor cells should be independent of cell cycle dynamics. Since this method does not distinguish between lineages derived from symmetric or asymmetric cell divisions, we limited our analysis to lineages with a maximum of 12 cells, the larger clonal size of neurogenic lineages in the *Emx1-CreER^{T2};MADM^{TG/GT}* experiments (clones with more than 12 cells account for less than 5% of the neurogenic lineages and largely include [87%] the outcome of symmetrically-dividing progenitor cells).

Consistent with the other approaches, the majority of lineages (~75%) labeled by injection of *Emx1-CreER^{T2};RCL-Gfp* mice with low tamoxifen doses at E12.5 were translaminar (*Figure 4b,e*). We also confirmed that ~13% of the lineages were restricted to deep cortical layers (*Figure 4c,e*). In addition, we found that ~11% of the lineages consist of PCs confined to superficial layers of the neocortex (*Figure 4d,e* and *Figure 4—figure supplement 1*). In sum, the combined results of three different sets of lineage-tracing experiments suggested that translaminar (~80%), deep layer-restricted (~10%) and superficial layer-restricted (~10%) lineages are generated at the onset of neurogenesis in the developing neocortex.

## Pyramidal cell lineages acquire diverse configurations

We next explored the precise organization of cortical lineages derived from VZ progenitor cells at E12.5. In lineage-tracing experiments using *Emx1-CreER^{T2};RCL-Gfp* mice (*Figure 5a*), we observed that only about a quarter of traced lineages contains neurons in every cortical layer from II to VI, and every other clone lacks neurons in one or multiple cortical layers (*Figure 5b–d,f*). For instance, a significant proportion of translaminar lineages lack PCs in layer V (*Figure 5b,f*) or layer IV (*Figure 5c,f*) but, considered collectively, PC lineages adopt every possible configuration of laminar distributions in the neocortex (*Figure 5f*). The heterogeneous organization of cortical lineages was not exclusively

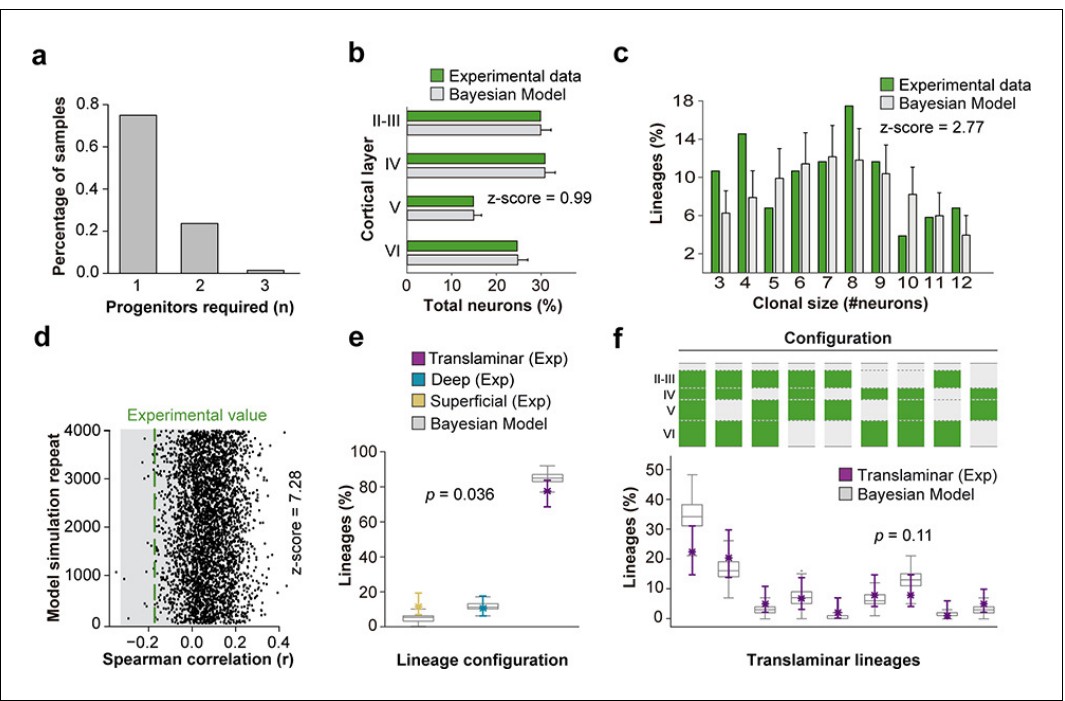

**Figure 6.** A stochastic model of cortical neurogenesis. (**a**) Number of progenitor identities required to reproduce experimental lineage configurations inferred by Bayesian modeling. The y axis represents the fraction of simulations from a total of 4000 that demand a particular minimum number of progenitors. (**b**) Fraction of cells in each cortical layer (expressed as percentage of total) in experimental and modeled lineages. (**c**) Clonal size distribution in experimental and modeled lineages. (**d**) Spearman correlation (r) values for the fraction of superficial and deep layer neurons in modeled lineages. Each dot represents an r value for one simulation. The green line shows the experimental value; the shadow area around the experimental data represents a 95% confidence interval for the experimental value. (**e**) Fraction of translaminar, deep and superficial layer-restricted lineages found experimentally and predicted by the model (expressed as percentage of all modeled lineages within a single simulation). Gray boxes represent variability among 4000 simulations; colored stars and lines show experimental values and 95% confidence intervals for experimental values (p=0.036, Chi-square test). (**f**) Relative frequency (expressed as percentage over all modeled translaminar lineages within a single simulation) of laminar configurations in experimental and modeled translaminar lineages. Gray boxes represent variability among 4000 simulations; colored stars and lines show experimental values and 95% confidence intervals for experimental values (p=0.11, Chi-square test). Histograms represent mean ± standard deviation. Z-scores represent the distance between experimental and simulated results for each parameter, which is calculated as the difference between the averages of model and experimental data divided by the standard deviation within model simulations (see Materials and methods for details). n = 103 neurogenic lineages in the primary somatosensory cortex of 25 animals. Data used for quantitative analyses as well as the numerical data that are represented in graphs are available in *Figure 4—source data 1* and *Figure 1—source data 2*. See also *Figure 6—figure supplement 1*. The online version of this article includes the following figure supplement(s) for figure 6:

**Figure supplement 1.** Laminar densities of pyramidal neurons do not predict lineage structure.

observed in the experiments performed in *Emx1-CreER^T2;RCL-Gfp* mice; similar results were obtained in retroviral lineage tracing (*Figure 5—figure supplement 1a–e*) and MADM experiments (*Figure 5—figure supplement 1f–j*). Although the clonal size of these lineages also exhibits great heterogeneity, it seems to follow a bimodal distribution, with maximums at approximately four and eight cells (*Figure 5e*). Intriguingly, these two maximums largely correspond to restricted and translaminar lineages respectively, reinforcing the previously described link between clonal size and laminar configuration.

We characterized the organization of translaminar lineages with PCs in every layer by quantifying the relative proportion of neurons in deep and superficial layers. This analysis revealed that these cortical lineages typically showed a bias toward the production of PCs for superficial layers, although a minority of lineages displayed a preference toward deep layers or a balanced distribution across

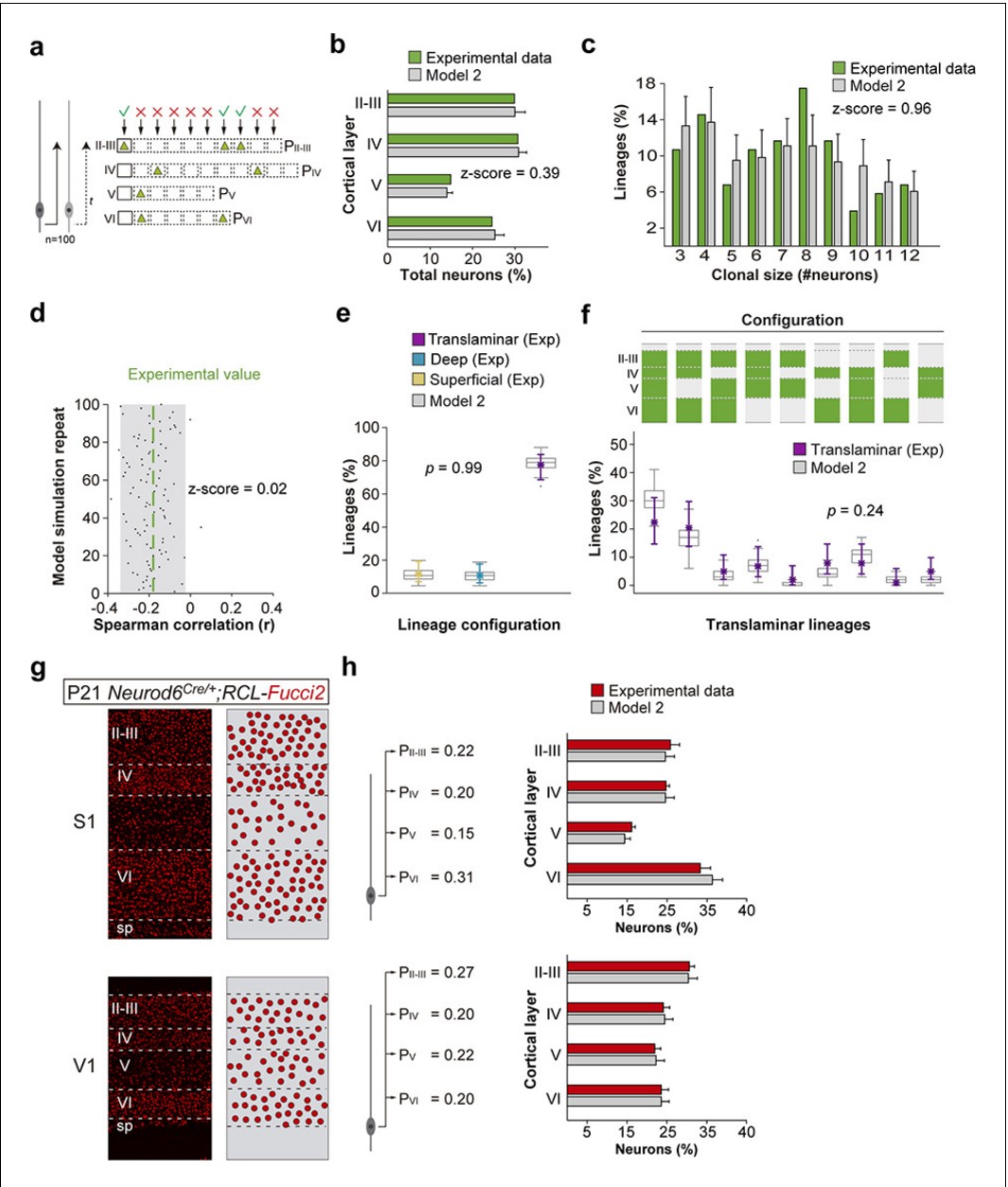

**Figure 7.** A small number of progenitor identities underlie lineage diversity. (**a**) Schematic representation of a mathematical model of cortical neurogenesis in which two different progenitor identities are modeled (Model 2). Squares represent the maximum number of stochastic decisions performed by each progenitor for each cortical layer during in silico simulations. The odds of generating neurons for each chance are given by a probability value (**P**), which is unique for each layer and progenitor identity. The model runs 100 simulations with 100 progenitors. (**b**) Fraction of cells in each cortical layer (expressed as percentage of total) in experimental and modeled lineages. (**c**) Clonal size distribution in experimental and modeled lineages. (**d**) Spearman correlation (r) values for the fraction of superficial and deep layer neurons in modeled lineages. Each dot represents an r value for one simulation. The green line shows the experimental value; the shadow area around the experimental data represents a 95% confidence interval for the experimental value. (**e**) Fraction of translaminar, deep and superficial layer-restricted lineages found experimentally and predicted by the model (expressed as percentage of all modeled lineages within a single simulation). Gray boxes represent variability among 100 simulations; colored stars and lines show experimental values and 95% confidence intervals for experimental values (p=0.99, Fisher's exact test). (**f**) Relative frequency (expressed as percentage over all modeled translaminar lineages within a single simulation) of laminar configurations in experimental and modeled translaminar lineages. Gray boxes represent variability among 100 simulations; colored stars and lines show experimental values and 95% confidence intervals

*Figure 7 continued on next page*

*Figure 7 continued*
for experimental values (p=0.24, Chi-square test). (**g**) 60 μm coronal sections through the primary somatosensory (S1) and visual (V1) cortex of P21 *Neurod6^{Cre/+};RCL*-Fucci2 mice. The schemas on the right illustrate PC densities per layer. (**h**) Fraction of PCs per layer (expressed as percentage of total neurons) generated with two sets of laminar probability factors using Model two compared to the experimental data. Histograms represent mean ± standard deviation. Z-scores represent the distance between experimental and simulated results for each parameter, which is calculated as the difference between the averages of model and experimental data divided by the standard deviation within model simulations (see Materials and methods for details). n = 103 neurogenic lineages in the primary somatosensory cortex of 25 animals. Data used for quantitative analyses as well as the numerical data that are represented in graphs are available in *Figure 4—source data 1*, *Figure 7—source data 1* and *Figure 1—source data 2*. The code for the generation of lineages can be found in *Figure 7—source code 1*. See also *Figure 7—figure supplement 1*.
The online version of this article includes the following source data, source code and figure supplement(s) for figure 7:

**Source data 1.** Laminar densities in *Neurod6^{Cre/+};RCL*-Fucci2 mice analyzed at P21.
**Source code 1.** Lineage generation simulator.
**Figure supplement 1.** Stochastic models considering single and multiple programs corroborate Bayesian inference.

deep- and superficial layers (*Figure 5g*). In general, the total amount of cells in superficial and deep cortical layers was slightly anti-correlated. To further explore the molecular diversity of PCs in these lineages, we stained P21 brain sections from *Emx1-CreER^{T2};RCL-Gfp* mice induced at E12.5 with antibodies against Ctip2 and Satb2, two transcription factors whose relative expression defines different types of PCs with unique patterns of axonal projections (*Greig et al., 2013*; *Lodato and Arlotta, 2015*). We identified four PC subclasses based on the expression of these markers and their laminar distribution (*Figure 5—figure supplement 2*): Cortico-cortical projection neurons (CCPN), subcerebral projection neurons (SCPN), cortico-thalamic projection neurons (CThPN) and heterogeneous projection neurons (HPN) (*Harb et al., 2016*). Using this classification, we found that nearly a quarter of all-layer translaminar lineages were composed exclusively by CCPNs, while multiple different combinations of PC identities comprise the remaining lineages (*Figure 5h*). Of note, only a minor fraction of all cortical lineages contains the entire complement of subtypes identified. Altogether, our experiments revealed that PC lineages exhibit a great degree of heterogeneity in the number and identities they comprise.

## Heterogeneous lineage configurations arise directly from neurogenesis

The observed heterogeneity in cortical lineages likely emerges during neurogenesis. However, it is possible that selective cell death of specific PCs might contribute to the heterogeneous organization of cortical lineages. Recent studies have shown PCs undergo apoptosis during early postnatal stages (*Blanquie et al., 2017*; *Wong et al., 2018*). To explore the contribution of cell death to the heterogenous configuration of cortical lineages, we labeled clones by injecting a low dose of tamoxifen in *Emx1-CreER^{T2};RCL-Gfp* mice at E12.5 and analyzed their laminar organization at P2, prior to the period of PC death (*Wong et al., 2018*). We detected no significant differences in the average clonal size or in the relative frequency of P2 translaminar and laminar-restricted lineages compared to P21 (*Figure 5—figure supplement 3a–d*). In addition, we observed that the diversity of lineage patterns was remarkably similar between P2 and P21 (*Figure 5—figure supplement 3e*). We also noticed a tendency ($\chi^2$ test, p=0.099) for the fraction of lineages with PCs in every layer to be larger and the frequency of lineages lacking PCs in layer five to be smaller at P2 compared to P21 (*Figure 5—figure supplement 3e*). These experiments suggested that although cell death may have a subtle impact in refining the final diversity of lineages and their relative proportion, such heterogeneity should arise directly during the process of cortical neurogenesis.

## Laminar densities do not predict lineage structure

The variability in size and composition of PC lineages raises questions about the developmental mechanisms underlying their genesis. We first asked whether lineage structure is relevant for cortical cytoarchitectural development. It is formally possible that the diversity in laminar composition of

cortical lineages is simply the consequence of a random process of PC generation in which the only boundary condition is the relative number of PCs that populate each layer of the cortex. To test this, we used the lineages mapped in the primary somatosensory cortex (S1) of *Emx1-CreER^{T2};RCL-Gfp* mice, which are meant to collectively generate a common pattern of laminar densities. We randomly permuted the PCs obtained from the different lineages while maintaining each neuron's laminar identity and the total number of cells in each lineage. If lineage structure were to exclusively affect the control of PC laminar fractions at population level, permuted lineages should be expected to match experimental data. As expected, the permutation process left unaltered the clonal size distribution and number of cells per layer observed in the experimental data (*Figure 6—figure supplement 1a,b*). However, it failed to replicate the observed anti-correlation in neuron numbers between superficial and deep layers (*Figure 6—figure supplement 1c*). In addition, we observed that the laminar configuration of the permuted lineages differed from the experimental data (data not shown). These results indicated that the laminar distribution of neurons within each lineage arises specifically from an organized pattern of neurogenesis.

## Stochastic models reproduce the diversity of progenitor outputs

One possibility to explain lineage diversity is the existence of multiple different VZ progenitor cell types with restricted potential to generate specific classes of PCs (*Franco and Müller, 2013*). However, the observed heterogeneity in lineage configurations may also arise from equipotent VZ progenitor cells that are subject to stochastic factors controlling their output, as proposed for the retina (*He et al., 2012*). To establish the feasibility of the latter scheme, we used a Bayesian approach to model the outcome of cortical progenitor cells following stochastic developmental programs (*Diana, 2019*; copy archived at https://github.com/elifesciences-publications/SampLin). This method involved using a set of probabilistic rules for generating lineages, and subsequently inferring the number of rules required for the assignment of all lineages observed in the experimental data (see Materials and methods for details). To avoid having to account for variability potentially attributable to differences in the distribution of lineages across areas of the neocortex, we only considered experimental data obtained from lineages mapped in S1 of *Emx1-CreER^{T2};RCL-Gfp* mice. We reasoned that some lineage configurations observed in our experiments (such those containing cells exclusively in deep cortical layers) could derive from an early interruption of the developing lineage, reflecting an early terminal division, or a progenitor cell undergoing cell death after a few rounds of division. Since the genesis of such lineages might therefore not arise from specific developmental programs, these configurations were also not taken into account for the inference of the stochastic rules governing this process. The Bayesian inference approach revealed that models using one or two progenitor types are sufficient to produce a diversity of lineage compositions as found in our experimental data (*Figure 6a,e,f*). However, the approach failed to reproduce the anti-correlation in cell numbers between superficial and deep layers found experimentally (*Figure 6d*) and tended to underestimate the fractions of superficial and small lineages in our data set (*Figure 6c,e*). This suggests that while simple stochastic processes acting mostly on a single homogeneous population of VZ progenitor cells can originate a vast diversity of outcomes as observed in our experiments, some experimental observations may arise from additional developmental programs and from features characteristic of the sequential process of neurogenesis.

## A small number of progenitor identities underlies lineage diversity

Having established that the stochastic behavior of a small number of progenitor types could in principle account for lineage diversity, we next explored specifically how diversity can result from the sequential dynamics of stochastic neuron generation. To this end, we simulated cortical progenitor behavior using models based on four basic rules derived from experimental knowledge (*Figure 7a* and *Figure 7—figure supplement 1a*). First, in silico progenitors would generate neurons for different layers sequentially, following the observed inside-out pattern. Second, each in silico progenitor would have a set, randomly selected number of opportunities to generate neurons in each layer. Third, for any progenitor, the decision to generate a neuron would be probabilistic, with cell generation probabilities varying by cortical layer but equal for all opportunities within the same layer. Thus, a progenitor type was defined by its specific combination of cell generation probabilities across layers. Fourth, to simulate the chances of premature terminal division and/or progenitor death, we

introduced a probabilistic chance of lineage interruption at each opportunity for cell generation. In silico lineages generated using this model were then compared with the experimental lineages mapped in the primary somatosensory cortex (S1) of *Emx1-CreER^T2;RCL-Gfp* mice. We set cell generation probabilities in each layer to match the total laminar fractions of PCs (*Figure 7b* and *Figure 7—figure supplement 1b*), as well as the clonal size distribution (*Figure 7c* and *Figure 7—figure supplement 1c*) observed in those lineages.

In agreement with the Bayesian approach, we found that a stochastic model based on a single, equipotent VZ progenitor cell (Model 1), that is a single set of cell generation probabilities, was able to reproduce the majority of experimentally observed lineage features. Modeled lineages recapitulated the existence of restricted lineages as well as all observed laminar configurations of translaminar lineages (*Figure 7—figure supplement 1e–f*). However, this model generated lineages with an exaggerated anti-correlation in the number of cells in superficial versus deep layers (*Figure 7—figure supplement 1d*) and failed to reproduce the bimodal distribution of clonal sizes, underestimating the fraction of small lineages (those containing 3–4 cells). In addition, the fraction of lineages restricted to superficial layers, which largely contribute to the small lineage sizes, was also underestimated (*Figure 7—figure supplement 1c,e*). These results suggested that the fraction of small superficial lineages is unlikely to arise from a single stochastic program common to all cortical progenitors.

We then generated a second model with two different sets of cell generation probabilities, defining two different progenitor populations (Model 2). In this model, the majority of progenitors belonged to a population generating the larger lineages, while a second, smaller population generated small progenies biased towards superficial layer PC fates (*Figure 7a*). We found that this model faithfully reproduced all the experimental features in our data: total laminar fractions (*Figure 7b*), bimodal distribution of clonal sizes (*Figure 7c*) and negative correlation in superficial versus deep layers (*Figure 7d*). In addition, the relative proportions of translaminar and laminar-restricted lineages were identical to those measured experimentally (*Figure 7e*). Finally, translaminar modeled lineages exhibited similar laminar configurations to the experimental lineages (*Figure 7f*). In sum, mathematical modeling suggests that a stochastic mechanism of cortical neurogenesis based on two independent progenitor cell populations best approximates the experimental data.

Finally, we explored whether the proposed stochastic model with two progenitor populations (Model 2) would be able to generate different ratios of layer-specific neurons under different circumstances (i.e. different cell generation probabilities), which would robustly account for the emergence of cytoarchitectural differences across neocortical areas. To this end, we quantified the fraction of PCs in each layer of the primary somatosensory (S1) and visual (V1) cortices in *Neurod6^Cre/+;*Fucci2 mice, in which all PCs in the neocortex are labeled with a nuclear fluorescent marker. As expected, we found important differences in laminar cytoarchitecture between both regions (*Figure 7g*). Remarkably, we found that subtle tuning of generation probabilities for both areas was sufficient to replicate the different laminar ratios in silico (*Figure 7h*). This suggests the stochastic mechanisms of neurogenesis described here would suffice to generate the diverse cytoarchitectonic patterns observed across neocortical areas.

## Discussion

Our results indicate that the output of individual progenitor cells in the developing mouse neocortex is much more heterogeneous than previously anticipated. Progenitor cells most frequently generate PCs for both deep and superficial layers of the neocortex, as suggested by previous studies. However, a sizable fraction of those lineages lacks PCs in one or several layers. In addition, the heterogeneous output of cortical progenitor cells includes lineages in which PCs are restricted to either deep or superficial layers. Mathematical modeling suggests that this wide diversity of outputs is compatible with a stochastic model of cortical neurogenesis. Such model represents a robust and adaptable mechanism for the assembly of neocortical cytoarchitecture.

### Methodological considerations

Understanding how individual lineages contribute to the production and organization of PCs is essential to articulate a coherent framework of cortical development. The analysis of the output of progenitor cells in the developing rodent cortex expands over three decades and has relied on four

approaches: retroviral labeling (*Luskin et al., 1988*; *Noctor et al., 2001*; *Noctor et al., 2004*; *Price and Thurlow, 1988*; *Reid et al., 1995*; *Walsh and Cepko, 1988*; *Walsh and Cepko, 1992*), mouse chimeras (*Tan et al., 1998*), MADM (*Beattie et al., 2017*; *Gao et al., 2014*) and genetic fate-mapping (*Eckler et al., 2015*; *Franco et al., 2012*; *García-Moreno and Molnár, 2015*; *Gil-Sanz et al., 2015*; *Guo et al., 2013*; *Kaplan et al., 2017*). These studies often led to contradictory results, which has prevented the emergence of a consistent model. The prevalent view is that each progenitor cell in the developing pallium is multipotent and generates a cohort of PCs that populate all layers of the neocortex except layer I (*Eckler et al., 2015*; *Gao et al., 2014*; *Guo et al., 2013*; *Kaplan et al., 2017*), as originally conceived in the radial unit hypothesis (*Rakic, 1988*). In contrast, some authors have suggested that many cortical progenitor cells are fate-restricted to generate PCs that exclusively occupy deep or superficial layers of the neocortex (*Franco et al., 2012*; *Franco and Müller, 2013*; *García-Moreno and Molnár, 2015*; *Gil-Sanz et al., 2015*). Here, we have used three different methods (retroviral labeling, MADM and genetic fate-mapping) to investigate the clonal production of cortical neurons by capitalizing on the synergy that emerges from the advantages of each individual approach. Our results indicate that this multi-modal approach is required to comprehensively capture the complex behavior of progenitor cells in the developing cortex.

Retroviral labeling has two important limitations: it only labels hemi-lineages and is prone to silencing, which may prevent the identification of the entire progeny of a progenitor cell (*Cepko et al., 2000*). Conversely, retroviral labeling targets progenitor cells indiscriminately and, consequently, is not biased toward a particular genetic fate (*Cepko et al., 2000*), as is the case for genetic strategies. MADM, on the other hand, has the enormous advantage of identifying both sister cells resulting from a cell division. However, G2-X MADM events require progenitor cells to undergo cell division at the time of induction because it directly relies on Cre-dependent inter-chromosomal mitotic recombination (*Zong et al., 2005*). Our results revealed that MADM does not reliably label a small fraction of progenitor cells present in the pallial VZ at E12.5 that gives rise to cohorts of PCs exclusively located in superficial layers of the neocortex. These lineages were however observed in both retroviral labeling experiments and in genetic tracing experiments using the same genetic driver (*Emx1-CreER$^{T2}$*) as in the MADM experiments, which strongly suggests that some Emx1+ progenitor cells producing exclusively superficial layer PCs in the developing cortex are not targeted by the MADM approach. We hypothesize that these progenitors might be quiescent or slow-dividing progenitors at this stage and become more active at later stages of development. Finally, although the use of genetic fate-mapping strategies (e.g. *Emx1-CreER$^{T2}$;RCL-Gfp*) is a powerful method to investigate cortical lineages, it has the important constraint of not being able to distinguish between symmetric proliferating and asymmetric neurogenic divisions. This hampers the analysis of clonal sizes, which can be otherwise accurately assessed with MADM except for the lineages that are not detected with this method.

## Diversity of neocortical lineages

Previous clonal analyses based on MADM lineage tracing experiments led to the suggestion that individual progenitor cells in the pallial VZ produce a unitary output of approximately eight excitatory neocortical neurons distributed throughout superficial and deep layers of the neocortex (*Gao et al., 2014*). However, those studies failed to identify lineages with restricted laminar patterns (either deep or superficial layer restricted clones). Consequently, they also underestimated the fraction of lineages with relatively small clonal size. In contrast, our analysis of neurogenic lineages revealed a bimodal distribution of clonal sizes with defined peaks centered at approximately four and eight cells, which largely correspond to the contribution of laminar-restricted and translaminar lineages, respectively.

Previous studies have suggested that some neocortical progenitor cells generate laminar-restricted lineages of PCs (*Franco et al., 2012*; *Gil-Sanz et al., 2015*). In our experiments, approximately one in six cortical progenitor cells generate laminar-restricted lineages. The existence of lineages restricted to deep layers of the neocortex was observed with all three methods used in this study. Although some variation exists in the relative fraction of deep layer-restricted lineages observed with the different approaches, these differences lie within the expected experimental noise considering the relatively small number of lineages that belong to this category. In addition, both retroviral labeling and genetic fate-mapping experiments identified a fraction of cortical progenitor cells that generate PCs that exclusively populate the superficial layers of the neocortex. It is

conceivable that these lineages reflect the output of progenitor cells that had already produced an earlier cohort of deep layer neurons prior labeling. If this where the case, one should expect to observe similar results in MADM experiments. In such experiments, however, we did not recover a significant fraction of superficial lineages. Therefore, the discrepancy between the results of genetic fate-mapping and MADM experiments, in which the same mouse strain is used as the driver for recombination (*Emx1-CreER$^{T2}$*), suggests that these fate-restricted lineages arise from progenitor cells that are not actively dividing at E12.5. This hypothesis is consistent with the identification of a population of self-renewing progenitors with limited neurogenic potential during the earliest phases of corticogenesis (*García-Moreno and Molnár, 2015*). The existence of superficial layer-restricted cortical lineages is further supported by the identification of IPCs as early as E12.5 that generate superficial layer PCs (this study and *Mihalas et al., 2016*), when the majority of deep layer PCs are being generated. Since IPCs derive from VZ progenitor cells (*Haubensak et al., 2004*; *Miyata et al., 2004*; *Noctor et al., 2004*) and cortical neurogenesis begins at these stages in the mouse, this observation reinforces the idea that some progenitors are tuned to generate superficial layer PCs from early stages of corticogenesis. Although it is formally possible that some translaminar lineages in the retroviral and genetic fate-mapping experiments could arise from symmetric divisions generating two laminar-restricted lineages (one superficial and one deep), this possibility is unlikely considering the results of the MADM experiments. The results of those experiments indicate that symmetric divisions at E12.5 generate at least one translaminar sub-lineage in virtually all cases (58/59), suggesting that most translaminar lineages arise from progressive divisions of individual progenitor cells.

Our study also revealed that, independently of the laminar distribution, individual cortical progenitor cells generate lineages with very diverse combinations of PC types. Cortical progenitors are thought to undergo progressive changes in their competency to generate different layer-specific types of PCs (*Desai and McConnell, 2000*; *Oberst et al., 2019*; *Rakic, 1974*). Consistent with this idea, our results reveal that most cortical progenitors generate diverse types of excitatory neurons. However, since many cortical progenitor cells fail to generate neurons for at least one layer of the neocortex, the majority of cortical lineages does not include the entire diversity of excitatory neurons. In other words, the fraction of individual cortical lineages that would be considered as 'canonical' – that is containing all three main classes of excitatory projection neurons (CCPN, SCPN and CThPN) – is significantly smaller than previously anticipated. Considering the variance in clonal size and lineage composition of neocortical lineages, our results indicate that cortical progenitor cells exhibit very heterogeneous patterns of neuronal generation and specification. This interpretation challenges the view that the neuronal output of RGCs is deterministic (*Gao et al., 2014*).

## A stochastic model of cortical neurogenesis

Our results indicate that stochastic developmental programs, in which cortical progenitors undergo a series of probability-based decisions for the generation of the different PC fates, are capable of generating the wide diversity of lineage configurations observed in our experiments. Therefore, in spite of the great diversity of configurations that exist among individual neocortical lineages, our results suggest that their genesis does not require a corresponding heterogeneity in VZ progenitors. The model proposed here is somewhat reminiscent of that described for the developing rodent and zebrafish retina (*Gomes et al., 2011*; *He et al., 2012*). In line with our findings, stochastic mechanisms based on a single set of probability rules explain the genesis of most, but not all neuronal types in the mammalian retina (*Gomes et al., 2011*).

It is presently unclear whether laminar-restricted lineages arise from a pool of progenitor cells separate from those generating translaminar lineages or should simply be considered as extreme examples of the enormous diversity of lineage configurations uncovered by our study. The generation of lineages restricted to deep layers might be due to premature terminal division or death of the progenitor cell (*Blaschke et al., 1996*; *Mihalas and Hevner, 2018*), as considered in our models, but the existence of superficial layer-restricted lineages is more difficult to explain. Moreover, our mathematical model best reproduces the complex cytoarchitecture of the neocortex when two distinct progenitor cell identities are considered. Previous studies have identified morphological heterogeneity among pallial VZ progenitor cells (*Gal et al., 2006*). However, there is limited evidence for important molecular differences among these cells (*Mizutani et al., 2007*; *Pollen et al., 2014*; *Telley et al., 2016*). In the absence of a definitive molecular signature, our results suggest that while

a homogeneous population of progenitor cells following a common developmental program explains most of the observed outcomes, it fails to generate the fraction of small superficial lineages observed in the experiments. The introduction of a second population of progenitor cells is required to reproduce these lineages. Although these findings might suggest the existence of a small fraction of cortical progenitors tuned to preferentially generate superficial lineages, it should not be taken as a definitive proof of fate-restriction in cortical progenitors. In our model, replicating the experimental data does not require such progenitors to be restricted, but simply biased toward generating superficial fates.

Our study suggests that progenitor cells in different cortical areas are likely constrained by different probabilistic rules, which would contribute to the generation of the diverse cytoarchitectonic patterns found across the neocortex. Although the number of lineages recovered from each cortical region is insufficient to provide conclusive evidence for major regional differences, lineages located in different cortical areas seem to exhibit features that reflect the local cytoarchitecture. For instance, lineages lacking layer IV neurons were abundantly found in the retrosplenial cingulate and motor cortices, where this layer is remarkably small. How and when stochastic neurogenic decisions are made remains to be elucidated, but they likely depend on the influence of extrinsic and intrinsic signals on parameters such as cell cycle length, the asymmetric inheritance of cell components, the generation of dividing (IPCs) versus postmitotic progeny, and the membrane potential of progenitor cells (*Haydar et al., 2003*; *Lange et al., 2009*; *Pilaz et al., 2009*; *Roccio et al., 2013*; *Vitali et al., 2018*; *Wang et al., 2009*). Local signals in different neocortical areas would contribute to the tuning of progenitor cell behaviors to output different cytoarchitectures without the requirement of regional-specific progenitor populations. Consequently, this model allows great flexibility in the generation of heterogeneous cortical cytoarchitectures without the requirement of a large number of progenitor identities. The specification of a very small number of progenitor cells with competence to adapt their neurogenic behavior to different probabilistic rules based on their location within the neocortical neuroepithelium represents the most parsimonious and robust mechanism for the generation of cortical circuitry.

# Materials and methods

**Key resources table**

| Reagent type (species) or resource | Designation | Source or reference | Identifiers | Additional information |
|---|---|---|---|---|
| Genetic reagent (*M. musculus*) | Neurod6$^{Cre}$ | PMID: 17146780 | RRID: MGI:4429427 | Dr Klaus Nave (MPI-EM) |
| Genetic reagent (*M. musculus*) | Emx1-Cre$^{ERT2}$ | PMID: 16388308 Jackson laboratory | Stock: 027784 RRID: IMSR_JAX:027784 | |
| Genetic reagent (*M. musculus*) | Tbr2$^{CreERT2}$ | PMID: 23897762 | RRID:MGI:5499789 | Dr Sebastian J Arnold (University of Freiburg) |
| Genetic reagent (*M. musculus*) | RCL-Gfp, RCE | PMID: 19363146 Jackson laboratory | Stock: 32037 RRID: MGI:4420759 | |
| Genetic reagent (*M. musculus*) | MADM$^{TG}$ | PMID: 21092859 Jackson laboratory | Stock: 013751 RRID:IMSR_JAX:013751 | |
| Genetic reagent (*M. musculus*) | MADM$^{GT}$ | PMID: 21092859 Jackson laboratory | Stock: 013749 RRID:IMSR_JAX:013749 | |
| Genetic reagent (*M. musculus*) | RCL-Fucci2a | PMID: 25486356 | RRID:IMSR_HAR:6899 | Dr Richard Mort (Lancaster University) |
| Cell line (*Homo sapiens*) | HEK293FT | Invitrogen | Cat: R700-07 RRID: CVCL_6911 | |
| Transfected construct (*M. musculus*) | rv::dio-eGFP | PMID: 23933753 Addgene | ID: 87662 | |

*Continued on next page*

Continued

| Reagent type (species) or resource | Designation | Source or reference | Identifiers | Additional information |
|---|---|---|---|---|
| Transfected construct (*M. musculus*) | *rv::dio-mCherry* | PMID: 23933753 Addgene | ID: 87664 | |
| Antibody | Anti-GFP (chicken polyclonal) | Aves Lab | Cat: GFP-1020 RRID: AB_10000240 | IF(1:2000) |
| Antibody | Anti-DsRed (rabbit polyclonal) | Clonetech | Cat: 632496 RRID: AB_10013483 | IF(1:500) |
| Antibody | Anti-mCherry (goat polyclonal) | Antibodies online | Cat: ABIN1440057 RRID: AB_11208222 | IF(1:500) |
| Antibody | Anti-Ctip2 (rat monoclonal) | Abcam | Cat: AB18465 RRID: AB_2064130 | IF(1:500) |
| Antibody | Anti-Satb2 (mouse monoclonal) | Abcam | Cat: AB51502 RRID: AB_882455 | IF(1:500) |
| Antibody | Anti-Satb2 (rabbit polyclonal) | Abcam | Cat: AB34735 RRID: AB_2301417 | IF(1:1000) |
| Antibody | Anti-Tle4 (goat polyclonal) | Gift from Dr Stefano Stifani (McGill University) | | IF(1:200) |
| Antibody | Anti-chicken IgY-Alexa fluor 488 (goat polyclonal) | TermoFisher | Cat: A-11039 RRID: AB_2534096 | IF(1:400) |
| Antibody | Anti-mouse IgG-Alexa fluor 647 (goat polyclonal) | TermoFisher | Cat: A-21240 RRID: AB_2535809 | IF(1:400) |
| Antibody | Anti-mouse IgG-biotinilated (Horse polyclonal) | Vector Labs | Cat: BA2000 RRID: AB_2313581 | IF(1:400) |
| Antibody | Anti-rat IgG-Alexa fluor 555 (goat polyclonal) | TermoFisher | Cat: A-21434 RRID: AB_2535855 | IF(1:400) |
| Antibody | Anti-goat IgG-Alexa fluor 555 (donkey polyclonal) | TermoFisher | Cat: A-21432 RRID: AB_2535853 | IF(1:400) |
| Antibody | Anti-rabbit IgG-Alexa fluor 488 (donkey polyclonal) | TermoFisher | Cat: A-21432 RRID: AB_2535853 | IF(1:400) |
| Chemical compound, drug | Tamoxifen | Sigma-Aldrich | Cat: 85256 | |
| Software, algorithm | Prism 8 | GraphPad | RRID: SCR_002798 | |
| Software, algorithm | MATLAB | Mathworks | RRID: SCR_001622 | |
| Software, algorithm | RStudio | RStudio | RRID: SCR_000432 | |
| Software, algorithm | Imaris 8 | Bitplane | RRID: SCR_007370 | |

## Mice

The following transgenic mouse lines were used in this study: *Neurod6$^{Cre}$* (*Goebbels et al., 2006*) RRID: MGI:4429427, *Emx1-Cre$^{ERT2}$* (*Kessaris et al., 2006*) RRID: IMSR_JAX:027784, *Tbr2$^{CreERT2}$* (*Pimeisl et al., 2013*) RRID: MGI:5499789, *RCL-Gfp* (*Sousa et al., 2009*) RRID: MGI:4420759, MADM$^{TG}$ (JAX 013751) RRID: IMSR_JAX:013751, MADM$^{GT}$ (JAX 013749) RRID: IMSR_JAX:013749 and *RCL*-Fucci2 RRID: IMSR_HAR:6899. The MADM$^{TG}$ and MADM$^{GT}$ alleles were generated by inserting $dT_{N-term}–Gfp_{C-term}$ and $Gfp_{N-term}–tdT_{C-term}$ sequences in the *Hipp11* locus (*Hippenmeyer et al., 2010*). *RCL*-Fucci2 are reporter mice in which a fluorescent ubiquitination-based cell cycle indicator (Fucci) consisting of *mVenus-hGem* and *mCherry-hCdt1* sequences linked by a T2A linker were flanked by loxP sited and inserted into the ROSA26 locus (*Mort et al., 2014*).

All adult mice were housed in groups and kept on reverse light/dark cycle (12/12 hr) regardless of genotypes. Only time-mated pregnant female mice that have undergone in utero surgeries were house individually. Both male and female mice were used in all experiments. In utero experiments where performed at different developmental stages that range from E9.5 to E14.5. For histological analyses, mice ages range from P2 to P30. All procedures were approved by King's College London and IST Austria, and were performed under UK Home Office project licenses, and in accordance with Austrian Federal Ministry of Science and Research license, and European regulations (EU directive 86/609, EU decree 2001–486). The day of vaginal plug was considered as embryonic day (E) 0.5 and the day of birth as postnatal day (P) 0.

## Retroviral infection for clonal labeling

Cre-dependent conditional retroviral stocks encoding EGFP and membrane-bound mCherry reporters (*rv::dio-eGfp* and *rv::dio-mCherry*) (*Ciceri et al., 2013*) were produced as previously described (*Tashiro et al., 2006*). In brief, Moloney murine leukemia viruses (MoMLV) were produced by transfecting HEK293T cells (RRID: CVCL_6911) with retroviral plasmids (*dio-eGfp* or *dio-mCherry*, *pCMV-Vsvg*, and *pCMV-GAG-pol*) using lipofectamine 2000. Forty-eight hours post-transfection, the supernatant was collected, concentrated and purified by two sequential rounds of ultracentrifugation. The viral pellet was re-suspended in sterile PBS and stored in aliquots at −80°C. Viral stocks for *dio-eGfp* and *dio-mCherry* were produced in the same plates and mixed before concentration by ultracentrifugation.

For in utero injections, pregnant females were deeply anesthetized with isoflurane and the abdominal cavity was incised to expose uterus. Conditional retroviruses were injected at low-titer into the telencephalic ventricles of E9.5, E10.5, E11.5, E12.5 and E14.5 mouse embryos using an ultrasound-guided imaging system (Visualsonic) coupled with a nanoliter injector as previously described (*Ciceri et al., 2013*; *Pla et al., 2006*). Some experiments were performed using the *rv::dio-eGfp* exclusively. After the procedure, the uterine horns were place back in the abdominal cavity and the wound was surgically sutured. The female was then placed in a 32°C recovering chamber for 30 mins post-surgery before returning to standard housing conditions.

## Inducible genetic clonal labeling

*Emx1-Cre$^{ERT2}$;RCL-Gfp* and *Tbr2$^{CreERT2/+}$;RCL-Gfp* pregnant females received a single intraperitoneal injection of low-dose (1 ng/kg) tamoxifen dissolved in corn oil at E12.5. MADM clones were generated as described previously (*Beattie et al., 2017*; *Hippenmeyer et al., 2010*). In brief, timed pregnant females were injected intraperitoneally with tamoxifen dissolved in corn oil at E12.5 at a dose of 2–3 mg/pregnant dam. Live embryos were recovered at E18–E19 through caesarean section, fostered, and raised for further analysis at P21.

## Histology

Postnatal mice were perfused transcardially with 4% paraformaldehyde (PFA) in PBS and the dissected brains were fixed for 2 hr at 4°C in the same solution. Brains were serially sectioned at 100 μm on a vibratome (VT1000S, Leica) or on a freezing microtome (SM 2010R, Leica) and free-floating coronal sections were then subsequently processed for immunohistochemistry as previously described (*Pla et al., 2006*).

The following primary and secondary antibodies where used: chicken anti-GFP (1:2000 Aves lab cat. no. GFP-1020, RRID:AB_10000240), rabbit anti-DsRed (1:500 Clonetech cat. no. 632496, RRID: AB_10013483), goat anti-mCherry (1:500 Antibodies-Online cat. no. ABIN1440057, RRID:AB_11208222), rat anti-Ctip2 (1:500 Abcam cat. no. Ab18465, RRID:AB_2064130), mouse anti-Sabt2 (1:500Abcam cat. no. Ab51502, RRID:AB_882455), rabbit anti-Sabt2 (1:1000 Abcam cat. no. Ab34735, RRID:AB_2301417), goat anti-Tle4 (1:200 gift from Stefano Stifani), anti-chicken IgY (H+L) 488 (1:400 Molecular Probes cat. no. A-11039, RRID:AB_2534096), anti-mouse IgG1 647 (1:400 Molecular Probes cat. no. A-21240, RRID:AB_2535809), anti-mouse IgG (H+L) biotinylated (1:400 Vector laboratories cat. no. BA-2000, RRID:AB_2313581), anti-rat IgG (H+L) 555 (1:400 Molecular Probes A-21434, RRID:AB_2535855), anti-goat IgG (H+L) 555 (1:400 Molecular Probes cat. no. A-21432, RRID:AB_2535853), and anti-rabbit 488 (1:400 Molecular Probes cat. no. A-21206, RRID: AB_2535853).

## Imaging

Images were acquired using fluorescence microscopes (DM5000B, CTR5000 and DMIRB from Leica or Apotome.2 from Zeiss) coupled to digital cameras (DC500 or DFC350FX, Leica; OrcaR2, Hamamatsu) with the appropriate emission filter sets or in inverted confocal microscopes (Leica TCS SP8 and Zeiss LSM800 Airyscan).

## In silico modeling of cortical development

All modeling of progenitor behavior was performed using MATLAB (MathWorks; RRID:SCR_001622). To avoid overfitting variability that could correspond to differences in progenitor behavior across cortical areas, simulations were compared to the lineages observed in primary somatosensory cortex (S1), using the *Emx1-CreER^T2^;RCL-Gfp* experiments. The structural similarity of the results of a model with the experimental data was assessed based on three parameters: proportion of cells per layer, clonal size distribution and Spearman correlation ($r$) values for number of cells in upper versus lower layers. For each parameter, we computed a normalized z-score measure by taking the difference between the experimental value and the average value across simulation repeats, and then dividing by the standard deviation across simulation repeats. Thus, z-score values over one would reflect a distance between experimental and modeled data larger than the standard deviation between simulation repeats.

To generate randomly permuted cortical lineages, neurons observed in the *Emx1-CreER^T2^;RCL-Gfp* experiments were permuted among lineages while maintaining their laminar identities. This operation was repeated 1000 times, providing average and standard deviation values that were then used to compare with the experimental results.

## Bayesian inference of progenitor types

To perform statistical inference on the number of categories required to explain the distribution of lineages throughout cortical layers, we employed a statistical model where $N$ observed lineages are grouped in $K$ progenitor types. Each type $t = 1, \ldots, K$ is associated to a vector of four probabilities $p_t = \{p_t^{(II/III)}, p_t^{(IV)}, p_t^{(V)}, p_t^{(VI)}\}$ representing the probabilities of any progenitor in the class to generate neurons in each of the four layers. We assume that each observed lineage can be assigned to a unique progenitor type based on its occupancy distribution. Progenitor types are associated with frequencies $f_t$, reflecting how likely a lineage is to belong to type $t$. The occupancy probabilities and the relative frequencies for each type as well as the number of types $K$ required can be obtained using Bayesian inference according to the Bayes' theorem

$$p(t_{1:N}, p_{1:K}, f_{1:K}|S) = \frac{\overbrace{P(t_{1:N}, S|p_{1:K}, f_{1:K})}^{likelihood} \cdot \overbrace{P(p_{1:K}; f_{1:K})}^{prior}}{\underbrace{P(S)}_{marginal\, likelihood}}$$

where $S$ is the count matrix whose elements $S_{ij}$ indicate how many neurons in lineage $i$ belong to layer $j$. The Bayes' theorem provides the posterior distribution of the model parameters $p$ and $f$ as well as the type assigned to each lineage conditional to the observations.

Our Bayesian model can be viewed as the following two-step generative process:

1. Each lineage $i$ is assigned to a progenitor type $t_i$ drawn independently from a categorical distribution with frequencies $f$.
2. The occupancy vector $S_{ij}$ of each lineage $i$ at each layer $j$ is drawn from a binomial distribution $Binomial\left(p_{t_i}^{(j)}; N_{max}\right)$ where $N_{max}=20$ is the maximum number of cells that can occupy each layer.

The likelihood of a given lineage assignment and count matrix can be written as

$$P(t_{1,\ldots,K}, S|p_{1,\ldots,K}, f_{1:K}) = \left(\prod_{t=1}^{K} f_t^{n_t}\right) \prod_{i=1}^{N} \prod_{j \in layers} \left[p_{t_i}^{S_{ij}} \left(1 - p_{t_i}^{N_{max}-S_{ij}}\right)\right]^{\sigma_{ij}},$$

where we introduced the binary variable $\sigma_{ij}$ to denote whether the matrix element $S_{ij}$ is included in the likelihood (in which case $\sigma_{ij} = 1$) or not ($\sigma_{ij} = 0$). In particular, the selection variable $\sigma_{ij}$ for each

lineage was set in such a way to exclude for each lineage the most superficial empty layers not followed by an occupied layer. The corresponding zero counts in the matrix *S* might be spurious due to external processes stopping the lineage at early stages.

To perform Bayesian inference, we used Dirichlet priors on the relative frequencies and Beta distributions as priors on the occupancy probabilities $p_{1,...,K}$. To draw samples of model parameters and progenitor types from the posterior distribution we implemented a Gibbs sampler (*Diana, 2019*) which combines data likelihood and prior distributions to explore the parameter space efficiently. In order to draw statistical samples of the number of classes *K*, we employed the Dirichlet process prior technique which allows us to remove existing classes or introduce new ones when assigning lineages to classes within the Gibbs sampler (*Diana, 2019*).

### Sequential stochastic modelling of lineage generation

Probabilistic models 1 and 2 simulated 100 progenitors undergoing cell generation sequentially, following the in vivo inside-out pattern (*Figure 7—source code 1*). In each layer, in silico progenitors took a number of stochastic decisions for neuron generation; at each decision, a new neuron could be generated, or alternatively, the chance could be skipped without neuron generation. Sequential generation of neurons thus used the following parameters. Number of opportunities per layer was set randomly and could vary between a minimum of one and a maximum equal to the maximum number of cells found for that layer in any single experimental lineage across our three types of experiments. This parameter establishes the number of stochastic decisions available to the progenitor and reflects the size of the temporal 'window' within which a progenitor can generate neurons for a given layer. Probability of cell generation, also layer-specific, gave the likelihood that a neuron is actually generated at each decision point. Simulations were repeated 100 times. Lineages smaller than three cells or larger than 12 cells were discarded from analysis.

For each model, the set of laminar division probabilities was adjusted to fit the experimental data regarding clonal size and laminar fractions of cells. Model 1 used a unique progenitor, that is a single set of laminar division probabilities. Model 2 incorporated an additional population and was fit by varying both the relative size of the two populations and the values of their division probabilities, including how probabilities varied across layers.

### Quantification and statistical analysis

*Cell distributions and clonal spatial configuration*. In all the experiments, brain sections were sequentially analyzed in rostral-to-caudal order and PC clones throughout the entire neocortex were identified as sparse, spatially separated cell clusters. The boundaries between cortical layers were traced based on nuclear (DAPI) staining and the laminar position of each cell was recorded accordingly. PC clones were classified as translaminar, infragranular and supragranular clones according to the laminar position of the neurons belonging to each clone. Cortical areas were identified based on the reference atlas of adult mouse brain (Allen Brain Atlas; http://www.brain-map.org). In *Emx1^CreER^*; MADM^TG/GT^ experiments, lineages derived from symmetric divisions (defined as lineages with three or more cells expressing each reporter) were excluded. In the *Emx1^CreER^;RCL-Gfp* experiments, lineages derived from symmetric divisions (defined as lineages containing more than 12 neurons) were excluded. Lineages containing one or two cells were also excluded in *Emx1^CreER^*;MADM^TG/GT^ and *Emx1^CreER^;RCL-Gfp* experiments. In retroviral experiments, one-cell and two-cell clones were considered separately in E9.5, E10.5, E11.5 and E12.5 experiments. In E14.5 experiments, two-cell lineages were considered, while one-cell clones were not quantified. This is due to the fact that, unlike at earlier time points, RGCs may be undergoing their last neurogenic division at this stage, and thus lineages with a minimum of two cells may be derived from targeted apical progenitor cells.

*Pyramidal cell types*. Brain sections were stained for markers of cortical projection neuron identity and classified based on the relative expression of the transcription factors Ctip2 and Satb2 in four main subtypes: Cortico-cortical (CCPN), sub-cerebral (SCPN), cortico-thalamic (CthPN) and heterogeneous (HPN) projection neurons. This last type was defined as layer V cells expressing both Ctip2 and Satb2 markers, which have been recently described as a distinct identity[33]. Images were captured using a confocal microscope and analyzed using a custom algorithm written in MATLAB (Mathworks). In brief, cell nuclei were segmented using the disk morphological function based on size and thresholds of fluorescence intensity over background. Cells were categorized as expressing high or

low levels of the transcription factors Ctip2 and Satb2 and further subclassified as CCPN (Ctip2$^{Low}$/Satb2$^{High}$), SCPN (Ctip2$^{High}$/Satb2$^{Low}$) or HPN (Ctip2$^{High}$/Satb2$^{High}$), based on the combination of marker expression. To distinguish between CThPN from CCPN in layer VI we used the following criteria: CCPN (Ctip2$^{Low}$/Satb2$^{High}$ OR Ctip2$^{Low}$/Satb2$^{Low}$) or CThPN (Ctip2$^{High}$/Satb2$^{Low}$). This allowed for the subclassification of layer V and layer VI cells based on the same set of markers. We verified these criteria by staining brain sections for the transcription factor Tle4 (*Figure 5—figure supplement 2*), a well-established specific marker of cortical CThPN identity (*Molyneaux et al., 2015*). Layer VI cells expressing high levels of both transcription factors were not classified, and lineages containing those cells were excluded from the quantification.

*Laminar ratios*. To quantify the actual densities of PCs in different cortical layers, *Neurod6$^{Cre}$* mice were crossed with Fucci2a reporter mice. The density of labeled red nuclei in each cortical layer was quantified from five representative serial sections of the somatosensory and visual cortex. Z-stacks were then 3d reconstructed and quantified using Imaris 8.1.2 (Bitplane; RRID:SCR_007370).

*Statistical tests*. Error bars in all graphs indicate standard deviation (std) unless otherwise stated in the legends. Comparisons of distributions over fractions of a total (e.g. *Figure 6e,f*) were analyzed using Fisher's exact test or Chi-square test. Average clonal size between lineages analyzed at P2 and P21 were analyzed using Mann-Whitney U-test. All statistical tests are specified in the figure legends.

## Acknowledgements

We thank I Andrew and SE Bae for excellent technical assistance, F Gage for plasmids, and K Nave for *Nex-Cre* mice. We thank members of the Marín and Rico laboratories for stimulating discussions and ideas. This work was supported by grants from the European Research Council (ERC-2017-AdG 787355 to OM and ERC-2016-CoG 725780 to SH). FKW was supported by an EMBO postdoctoral fellowship and is currently a Marie Skłodowska-Curie Fellow from the European Commission under the H2020 Programme, RB received support from FWF Lise-Meitner program (M 2416), and MM received support from the UKRI Medical Research Council (MR/P006639/1).

## Additional information

### Funding

| Funder | Grant reference number | Author |
|---|---|---|
| H2020 European Research Council | ERC-2017-AdG 787355 | Oscar Marin |
| H2020 European Research Council | ERC-2016-CoG 725780 | Simon Hippenmeyer |
| European Molecular Biology Organization | | Fong Kuan Wong |
| H2020 Marie Skłodowska-Curie Actions | | Fong Kuan Wong |
| Austrian Science Fund | Lise-Meitner program M 2416 | Robert Beattie |
| Medical Research Council | MR/P006639/1 | Miguel Maravall |

The funders had no role in study design, data collection and interpretation, or the decision to submit the work for publication.

### Author contributions

Alfredo Llorca, Gabriele Ciceri, Conceptualization, Formal analysis, Investigation, Visualization, Writing—original draft; Robert Beattie, Fong Kuan Wong, Formal analysis, Investigation; Giovanni Diana, Software, Formal analysis, Investigation, Visualization; Eleni Serafeimidou-Pouliou, Marian Fernández-Otero, Carmen Streicher, Investigation; Sebastian J Arnold, Resources, Writing—review and editing; Martin Meyer, Resources, Supervision, Writing—review and editing; Simon Hippenmeyer,

Supervision, Writing—review and editing; Miguel Maravall, Conceptualization, Software, Formal analysis, Visualization, Writing—review and editing; Oscar Marin, Conceptualization, Supervision, Funding acquisition, Project administration, Writing—review and editing

## Author ORCIDs
Alfredo Llorca (ID) https://orcid.org/0000-0001-5555-2839
Giovanni Diana (ID) http://orcid.org/0000-0001-7497-5271
Simon Hippenmeyer (ID) http://orcid.org/0000-0003-2279-1061
Miguel Maravall (ID) https://orcid.org/0000-0002-8869-7206
Oscar Marin (ID) https://orcid.org/0000-0001-6264-7027

## Ethics
Animal experimentation: All procedures were approved by King's College London and IST Austria, and were performed under UK Home Office project licenses, and in accordance with Austrian Federal Ministry of Science and Research license, and European regulations (EU directive 86/609, EU decree 2001- 486).

## Decision letter and Author response
Decision letter https://doi.org/10.7554/eLife.51381.sa1
Author response https://doi.org/10.7554/eLife.51381.sa2

## Additional files

### Supplementary files
• Transparent reporting form

### Data availability
All data generated or analysed during this study are included in the manuscript and supporting files.

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
