## [Decision Letter]

**Acceptance summary:**

Collectively, your study constitutes an impressive and highly convincing study showing for the first time the nuanced heterogeneity of cortical neurogenesis. By combining experimental approaches that are usually performed alone with a careful quantitative analysis and mathematical modeling, this works displays a clear picture of cortical neurogenesis which has been missing in the field. More broadly, it highlights the need to assess population levels to understand the full diversity of cell behavior and their collective output and that stochasticity is a major – yet understudied – factor in biological processes.

**Decision letter after peer review:**

Thank you for submitting your article "A stochastic framework of neurogenesis underlies the assembly of neocortical cytoarchitectures" for consideration by *eLife*. Your article has been reviewed by three peer reviewers, one of whom is a member of our Board of Reviewing Editors, and the evaluation has been overseen by Eve Marder as the Senior Editor. The following individual involved in review of your submission has agreed to reveal their identity: Denis Jabaudon (Reviewer #2).

Reviewers and myself as a reviewing editor found your study on the neuronal output of individual cortical progenitors "impressive", "interesting", and "an important contribution to the debate on fate-restricted progenitors and the origin of cortical pyramidal neuron diversity". They all outlined the "quality of the experimental work", of the quantifications, the importance of comparing in depth the three tracing methods with large samples, the interest of the modelisation and the conclusions emerging from this work.

They furthermore outline the importance of this study for the field and enthusiastically support publication in *ELife* without further experimentation. Nevertheless, it would be important to address the reviewer's comments in best possible ways. Access to the raw data (in Excel form) should be provided, as important in the context of the points raised by reviewer # 3.

Reviewer #1:

In this manuscript by llorca et al., the authors perform in-depth clonal analyses to assess the behavior of neocortical progenitors. This topic has been the focus of a major and yet unresolved controversy. On the one hand, most studies revealed that single progenitors give rise to an entire cortical column. On the other hand, subset of studies proposed that subsets of progenitors would specifically give rise to deep or upper layer clones. Here, by using an elegant combination of all the methods available (retroviral clonal analyses, MADM and low-dose of tamoxifen exposure), the authors reveal that distinct methods can induce biases. They also show a convincingly quantitative a careful cross-comparison using the three methods and reveal that while most progenitors produce translaminar, some produce deep-layer restricted clones and a small proportion upper layer-restricted clones. The authors furthermore highlight that translaminar clones do not include all the canonical cell types and that this heterogeneity is due to neurogenesis and not cell death. Finally, by performing mathematical modeling, they put show that a limited heterogeneity of progenitors and a stochastic output of neurogenesis can account for the results observed in vivo.

Reviewer #2:

In this study, Llorca et al. used 3 distinct approaches to address the neuronal output of individual progenitor cells, focusing on their laminar position. They report that a high fraction of clones displays translaminar cell positions at P21, while some clones are restricted to deep or superficial layers. In an attempt to identify the rules which govern the biological patterns they observe, the authors build two mathematical models: they find that probabilistic decisions in a restricted number of progenitor cell subtypes are in principle sufficient to account for the reported clonal laminar diversity. This work is well conducted and interesting. It is an important contribution to the debate on fate-restricted progenitors and the origin of cortical pyramidal neuron diversity.

1) Both in the title ("cytoarchitecture*s*"), the Abstract and the Introduction, the authors highlight that their findings explain how heterogeneous laminar organizations can be generated across cortical areas. While the second model they use indeed suggests that rules identified in S1 may apply in V1, the data provided are not sufficient to support the claims related to inter-areal diversity. Also, stochastic control over cell fate is not a "novel mechanism" and has been described in multiple settings. These elements of the text should be corrected to better reflect available data.

2) It would be very interesting to perform an analysis based on the radial position of cells in a clone rather than on their laminar position. Indeed the division of clones into "deep", "superficial" and "translaminar" is arbitrary and whether a cluster analysis of the clones would unbiasedly reveal such categories is unclear. This would allow for normalized inter-areal comparisons in clone behavior, and assess the extent to which laminar identity drives clonal distribution.

3) The authors should provide an annotated Excel table containing the laminar position (and, ideally, the radial position, see point 2 above) for each of the neurons of every clone used for the study. This manuscript represents an impressive amount of work and providing this raw data would add a remarkable "database" component to the existing findings.

Reviewer #3:

In this stimulating paper, the Marin laboratory has used a combination of three lineage tracing approaches to explore the mechanisms by which progenitor cells in the cerebral cortex generates the diversity of pyramidal cells (PCs) that are found in different cortical cell layers. The manuscript directly addresses a central question in the field that has been a matter of discussion. The classic model of neurogenesis suggests that a single progenitor generates in sequential order all subtypes of neurons. An alternative view suggests that there might be more than one progenitor type. Significantly, the data presented in the manuscript do not support the view that single progenitors generate in predefined sequential order different subtypes of neurons. Instead, the data support the existence of more than one progenitor and a stochastic model by which these progenitors generate subtypes of neurons.

I find the study interesting and important. A particular strength of the manuscript is the systematic analysis of lineage trees using three different approaches that are commonly used to study lineage relationship. Each of the methods has limitations, but their direct comparison in one experimental study mitigates the resulting complexity in data interpretation. The data in the paper are of high quality and well presented, and the conclusions are well supported by the experimental evidence.

I have several comments for consideration.

1) The overall conclusion of the retrovirus lineage tracing is sound, but there are some limitations that could be mentioned. First, retroviruses are injected for lineage tracing at E12.5. It is conceivable that some lineage restriction of progenitors could occur shortly after E12.5. This would be missed in the current analysis. Second, the authors assume that none of the mapped lineages go back to self-renewing progenitor divisions. It is formally possible that some retroviruses infect a self-renewing progenitor that in the next division produced two progenitors, each of which is lineage restricted. One of these progenitors could generate (for example) layer 5-6 neurons, one layer 2-4 neurons, or other layer combinations. This pattern could still be compatible with a clone size of 6-8 offspring. Overall, these caveats indicate that the study provides a minimal estimate of the number of lineage restricted progenitors.

2) Similar limitations apply to the Cre lineage tracing studies.

3) The data in the paper are not consistent with the conclusions reached in the paper by Gao et al., 2014, which proposed a deterministic model of progenitor behavior based on MADM studies. The current paper seems to avoid to clearly spell out that there is a serious discrepancy between the two papers. There are two major problems. First, the Gao paper and the current paper used the same MADM strategy for lineage studies, but they got different results. Second, the additional strategies used in the current paper are inconsistent with the conclusions reached by the previously reported MADM results. I will address this in more detail:

i) MADM strategy: in the previous paper, 2% of clones or less were restricted at E12-E13 to deep layers, and 10% were restricted to upper layers. In the current paper, 7% of the clones were restricted to deep layers and none to upper layers. While the numbers seem small, the fold difference is large. It might reflect the limitations of the MADM strategy. MADM experiments are complex and involve elaborate crossing schemes, data collection strategies and interpretation. Thus, small variations in experiments (for example a slight difference in the timing of Cre) significantly can affect the outcome of the experiments. MADM might capture major events but miss important pieces of the overall picture.

The Gao paper concluded that progenitors deterministically and sequentially generated neuronal subtypes (on average 8) for all neuronal layers. Data in Figure 5—figure supplement 1J directly contradict these data and demonstrate that MADM clones do not span all layers.

ii) Retroviruses and Cre strategies: the results obtained with retrovirus lineage tracing and Cre strategies are remarkably consistent, and they contradict the conclusions previously reached by Gao et al. The data show a wide variation in clonal size (Figure 5E, Figure 5—figure supplement 1I). Furthermore, like the MADM studies, they provide evidence that single progenitors generate neurons for different layers, but the clones do not cover all layers (Figure 5F). The data also show that a significant number of progenitors appear to produce a similar subtype of projection neurons (e.g. CCPNs) even if they are in different layers (Figure 5H), suggesting some kind of restriction in the fate potential of progenitors.

These are major discrepancies to the Gao study. I think it is important to discuss this clearly and to point out that the current data contradict the conclusions reached by Gao.

---

## [Author Response]

Reviewer #2:[…] 1) Both in the title ("cytoarchitectures"), the Abstract and the Introduction, the authors highlight that their findings explain how heterogeneous laminar organizations can be generated across cortical areas. While the second model they use indeed suggests that rules identified in S1 may apply in V1, the data provided are not sufficient to support the claims related to inter-areal diversity. Also, stochastic control over cell fate is not a "novel mechanism" and has been described in multiple settings. These elements of the text should be corrected to better reflect available data.

We agree that our data is insufficient to provide conclusive evidence regarding the generation of region-specific laminar ratios of PCs. Our mathematical model, however, is capable to explain the genesis of different regional cytoarchitectures using a stochastic program. This suggests that the origin of region-specific lamination may not require region specific molecular programs, but rather the precise tuning of a general stochastic mechanism. Nevertheless, following the suggestion of the reviewer we have edited the Title, Abstract and Introduction to avoid any confusion regarding this matter. Following the reviewer’s advice, we also avoided the use of the word “novel” when referring to the stochastic mechanisms described in the manuscript.

Actions taken:

- We have modified the text in the Title, Abstract and Introduction to address the reviewer’s concern.

- We have removed the word “novel” from the Abstract.

2) It would be very interesting to perform an analysis based on the radial position of cells in a clone rather than on their laminar position. Indeed the division of clones into "deep", "superficial" and "translaminar" is arbitrary and whether a cluster analysis of the clones would unbiasedly reveal such categories is unclear. This would allow for normalized inter-areal comparisons in clone behavior, and assess the extent to which laminar identity drives clonal distribution.

Although the proposed idea is certainly interesting, such analysis would require the re-annotation of all the clones based on the distance of the neurons from the pial or ventricular surfaces. This would need to be done manually and will likely take several months. Consequently, we believe this is out of the scope of our present study. In our study, laminar identity is a proxy (which we know it is imperfect) of neuronal identity. In addition, we use common molecular markers to identify type identity beyond laminar fates. Therefore, the specific radial position of the observed cells was not included in our original analysis since it does not provide additional information about cell fate.

3) The authors should provide an annotated Excel table containing the laminar position (and, ideally, the radial position, see point 2 above) for each of the neurons of every clone used for the study. This manuscript represents an impressive amount of work and providing this raw data would add a remarkable "database" component to the existing findings.

We thank the reviewer for his kind consideration of the amount of data reported in this manuscript. The revised version of this manuscript include annotated source files, as requested by the reviewer and consistent with *eLife* editorial recommendations.

Action taken:

- Raw data for which the analyses described in the manuscript are derived are now provided in Excel format as source files.

Reviewer #3:[…] I have several comments for consideration.1) The overall conclusion of the retrovirus lineage tracing is sound, but there are some limitations that could be mentioned. First, retroviruses are injected for lineage tracing at E12.5. It is conceivable that some lineage restriction of progenitors could occur shortly after E12.5. This would be missed in the current analysis. Second, the authors assume that none of the mapped lineages go back to self-renewing progenitor divisions. It is formally possible that some retroviruses infect a self-renewing progenitor that in the next division produced two progenitors, each of which is lineage restricted. One of these progenitors could generate (for example) layer 5-6 neurons, one layer 2-4 neurons, or other layer combinations. This pattern could still be compatible with a clone size of 6-8 offspring. Overall, these caveats indicate that the study provides a minimal estimate of the number of lineage restricted progenitors.2) Similar limitations apply to the Cre lineage tracing studies.

We thank the reviewer for pointing this interesting possibility. It is true that fate restriction could happen shortly after E12.5, and this vision is compatible with bot retroviral and Cre fate-mapping results. However, the MADM dataset is not consistent with this view. If two complementary fate-restricted progenitors were to arise from E12.5 progenitor divisions, we should have recovered an important fraction of symmetric lineages containing two laminar-restricted sub-lineages larger than two cells. This configuration is virtually absent from the MADM dataset, so we think this possibility is unlikely. We have now addressed this point in the Discussion.

Action taken:

- We have now referred to this issue in the Discussion under the heading “Diversity of neocortical lineages”.

3) The data in the paper are not consistent with the conclusions reached in the paper by Gao et al., 2014, which proposed a deterministic model of progenitor behavior based on MADM studies. The current paper seems to avoid to clearly spell out that there is a serious discrepancy between the two papers. There are two major problems. First, the Gao paper and the current paper used the same MADM strategy for lineage studies, but they got different results. Second, the additional strategies used in the current paper are inconsistent with the conclusions reached by the previously reported MADM results. I will address this in more detail:i) MADM strategy: in the previous paper, 2% of clones or less were restricted at E12-E13 to deep layers, and 10% were restricted to upper layers. In the current paper, 7% of the clones were restricted to deep layers and none to upper layers. While the numbers seem small, the fold difference is large. It might reflect the limitations of the MADM strategy. MADM experiments are complex and involve elaborate crossing schemes, data collection strategies and interpretation. Thus, small variations in experiments (for example a slight difference in the timing of Cre) significantly can affect the outcome of the experiments. MADM might capture major events but miss important pieces of the overall picture.The Gao paper concluded that progenitors deterministically and sequentially generated neuronal subtypes (on average 8) for all neuronal layers. Data in Figure 5—figure supplement 1J directly contradict these data and demonstrate that MADM clones do not span all layers.ii) Retroviruses and Cre strategies: the results obtained with retrovirus lineage tracing and Cre strategies are remarkably consistent, and they contradict the conclusions previously reached by Gao et al. The data show a wide variation in clonal size (Figure 5E, Figure 5—figure supplement 1I). Furthermore, like the MADM studies, they provide evidence that single progenitors generate neurons for different layers, but the clones do not cover all layers (Figure 5F). The data also show that a significant number of progenitors appear to produce a similar subtype of projection neurons (e.g. CCPNs) even if they are in different layers (Figure 5H), suggesting some kind of restriction in the fate potential of progenitors.These are major discrepancies to the Gao study. I think it is important to discuss this clearly and to point out that the current data contradict the conclusions reached by Gao.

We agree with the reviewer that our study reports important discrepancies with Gao et al., 2014. In addition to the matters pointed by the reviewer, superficial restricted lineages are not reported by Gao and colleagues when labelling at E12.5. We believe that the reason for the observed discrepancies are mainly technical, as discussed in the text. Another major discrepancy with the cited paper is the interpretation of progenitor output as “deterministic” and “predictable” considering the huge heterogeneity described in our manuscript. We have now more clearly stated these discrepancies in the Discussion.

Action taken:

- A sentence has been added in the Discussion that highlights the main discrepancy of our study with Gao et al., 2014.